# Dynamic changes to signal allocation rules in response to variable social environments in house mice

Caitlin H. Miller [1✉], Matthew F. Hillock [1], Jay Yang[1], Brandon Carlson-Clarke[1], Klaudio Haxhillari[1], Annie Y. Lee[1], Melissa R. Warden [1] & Michael J. Sheehan [1✉]

Urine marking is central to mouse social behavior. Males use depletable and costly urine marks in intrasexual competition and mate attraction. We investigate how males alter signaling decisions across variable social landscapes using thermal imaging to capture spatio-temporal marking data. Thermal recording reveals fine-scale adjustments in urinary motor patterns in response to competition and social odors. Males demonstrate striking winner-loser effects in scent mark allocation effort and timing. Competitive experience primes temporal features of marking and modulates responses to scent familiarity. Males adjust signaling effort, mark latency, and marking rhythm, depending on the scent identities in the environment. Notably, recent contest outcome affects how males respond to familiar and unfamiliar urine. Winners increase marking effort toward unfamiliar relative to familiar male scents, whereas losers reduce marking effort to unfamiliar but increase to familiar rival scents. All males adjust their scent mark timing after a contest regardless of fight outcome, and deposit marks in more rapid bursts during marking bouts. In contrast to this dynamism, initial signal investment predicts aspects of scent marking days later, revealing the possibility of alternative marking strategies among competitive males. These data show that mice flexibly update their signaling decisions in response to changing social landscapes.

[1] Department of Neurobiology and Behavior, Cornell University, Ithaca, NY, USA. ✉email: chm79@cornell.edu; msheehan@cornell.edu

Animals adjust their signaling behavior in response to recent experience and social context. Signalers may adjust not only the frequency of signaling behavior, but also when, where, and how they signal in response to changing social and physical environments[1–4]. In house mice (*Mus musculus domesticus*), males use metabolically costly urine marks to mediate intrasexual competition and mate attraction[5–10]. The abundance, spatial distribution, and chemical composition of urine marks contain information about a male's competitive status and identity[5–8,11–16]. While urine marks convey rich social information, they are also directly depletable. Just as a car runs out of fuel, animals have a limited supply of urine to allocate at any given moment. As a result, the timing of urine deposition is likely a crucial feature of scent-mark signaling. Here, we explore the flexibility of signal allocation decisions, both on a moment-to-moment timescale as well as over the course of days.

Male social relationships are shaped by competition and familiarity with conspecifics in house mice[5,17–23]. Urine marking mediates some of these relationships by allowing assessment and recognition of individuals[7,12,13,16,24]. Both stimulus familiarity and aggressive contests independently have strong effects on male urine, however it remains poorly understood how the two interact. In many territorial species, familiar neighbors reduce aggressive behaviors and signaling effort toward each other in order to lessen the costs of territorial defense, also known as the "dear enemy" effect[25–29]. Given the high costs and depletable nature of urine marks, males should dynamically modulate signal allocation as the landscape is updated with new social information. The present study aims to shed light on these decision rules by exploring how established competitive relationships and familiarity influence male signal allocation across social and scent-marked environments. The ability to keep track of experiences with specific individuals and respond to unfamiliar competitors is likely highly adaptive.

We investigate how males shift their signal allocation after an aggressive contest in response to the presence of a familiar male competitor, as well as to the presence of urine scent-marks of differing male identities. The objectives of this study were to: (1) implement thermal recording as a method for measuring scent marking in social contexts, (2) examine how competitive experience alters marking behavior, and (3) test the hypothesis that familiarity is important for signal allocation decisions. To do this, we developed a 4-day trial design in which 31 pairs of age and weight-matched breeding male house mice of two distinct wild-derived strains were paired as competitors and presented a series of social and scent-marked trials (Fig. 1a and Videos S1, S2). On the first day, paired males were placed in an arena separated by a mesh barrier (Fig. 1a and Video S1). Paired males could see, hear, and smell each other but were limited to minimal physical contact through the mesh. The mesh barrier was subsequently removed, and males engaged in an aggressive contest or "fight trial" (Fig. 1a). Based on the total aggressive behaviors performed by each male, males were unambiguously classified as winners or losers (Fig. S1 and Table S1). On the second day, each male was placed in an empty arena (Fig. 1a). On the third day, males were placed back into the mesh arena with the same male competitor they encountered on the first day (Fig. 1a). Finally, on the fourth day, each male was exposed to one of four urine-marked treatments. Each treatment contained aliquoted male urine of three possible identities (self, familiar male, or unfamiliar male) in two spatially distinct scent-marked zones (Fig. 1a and Video S2). The four treatment types span a range of scent-mark combinations (self-self, self-familiar, self-unfamiliar, familiar-unfamiliar), in which the familiar stimulus is the urine of a male's paired competitor and the unfamiliar stimulus is novel male urine of a third distinct

genotype (Fig. 1a). Urine marking and space use data were collected for each male across urine marking assays while aggression was scored in the fight trials (Figs. 1b, c, S1).

## Results

**Thermal imaging reveals spatiotemporal dynamics of scent marking in real time**. To fully understand urine allocation decisions in mice, we need to measure real-time spatial and temporal patterns of scent-mark deposition events. Mouse urine marking has previously been studied by capturing snapshots of marking patterns. More recently, thermal recording has been used to detect the voiding of urine in non-social contexts[5,13,30–33]. Urine leaves the body hot (close to body temperature) and quickly cools below the ambient substrate temperature, providing a distinctive thermal signature. Here, we used thermal imaging as an unobtrusive method for capturing the spatial and temporal allocation of urine marks by male house mice across social contexts (Fig. 1). Trials were performed on filter paper to present urine stimuli and to generate images of urine blots under UV light (Fig. 1d, e). This allowed us to compare thermal recording with a traditional urine detection method.

Using thermal imaging we recorded a total of 9,314 urine deposition events across trials and explored the temporal distribution of these depositions. We observed an initial spike in urine deposition with a peak of activity at ~100 s, followed by an exponential decline (Fig. 1f). The majority (77%) of marks are deposited within the first 15 min (800 s), suggesting males rapidly scent-mark upon entering an environment (Fig. 1f). Thermal imaging focuses on urine deposition, as marks are scored by the distinct thermal profile of urine as it is deposited. UV light imaging cannot distinguish between deposition and distribution events, as urine is further distributed by males tracking urine with their paws and tail. Additionally, urine deposited in close spatial proximity to existing marks can appear as a single mark under UV light at the end of a trial. The number of marks detected by thermal imaging and UV imaging did not differ significantly (M1: $F_{1,430} = 0.0034$, $p = 0.95$; Fig. 1g and Table S1). The two detection methods are also highly correlated (Fig. S2), justifying the use of thermal imaging to examine how temporal urine allocation varies across social contexts. The implementation of thermal recording in social assays opens new investigative avenues in social neuroscience, and insights into the neurophysiological basis of voluntary urination.

**Competitive experience and initial signal investment shape urine mark allocation**. Competitive social encounters can have a range of important consequences on the behavior and physiology of individuals. How individuals respond to contest outcomes is often dependent on the assessment of their resource holding potential[34–36]. Signals play a key role in such encounters as they can convey information about the competitive ability of individuals[37–39]. In house mice, the initial marking levels of males have been shown to contain information about their competitive ability[7]. We predicted that (1) higher-marking males would be more likely to win aggressive contests, (2) winners would increase while losers would decrease in signaling effort after a fight, and (3) temporal marking dynamics would be shaped by recent social experience. We compared urine marks in the presence of the same competitor before (Mesh 1) and after (Mesh 2) a fight (Fig. 1a). Fight outcome has a strong effect on the total number of urine marks (M2: $F_{1,68} = 10$, $p = 0.002$; Table S2), and there is a significant interaction between fight outcome and trial (M2: $F_{1,60} = 12$, $p = 0.001$; Table S2). Before the fight (Mesh 1), the to-be winners include more high-marking individuals than the

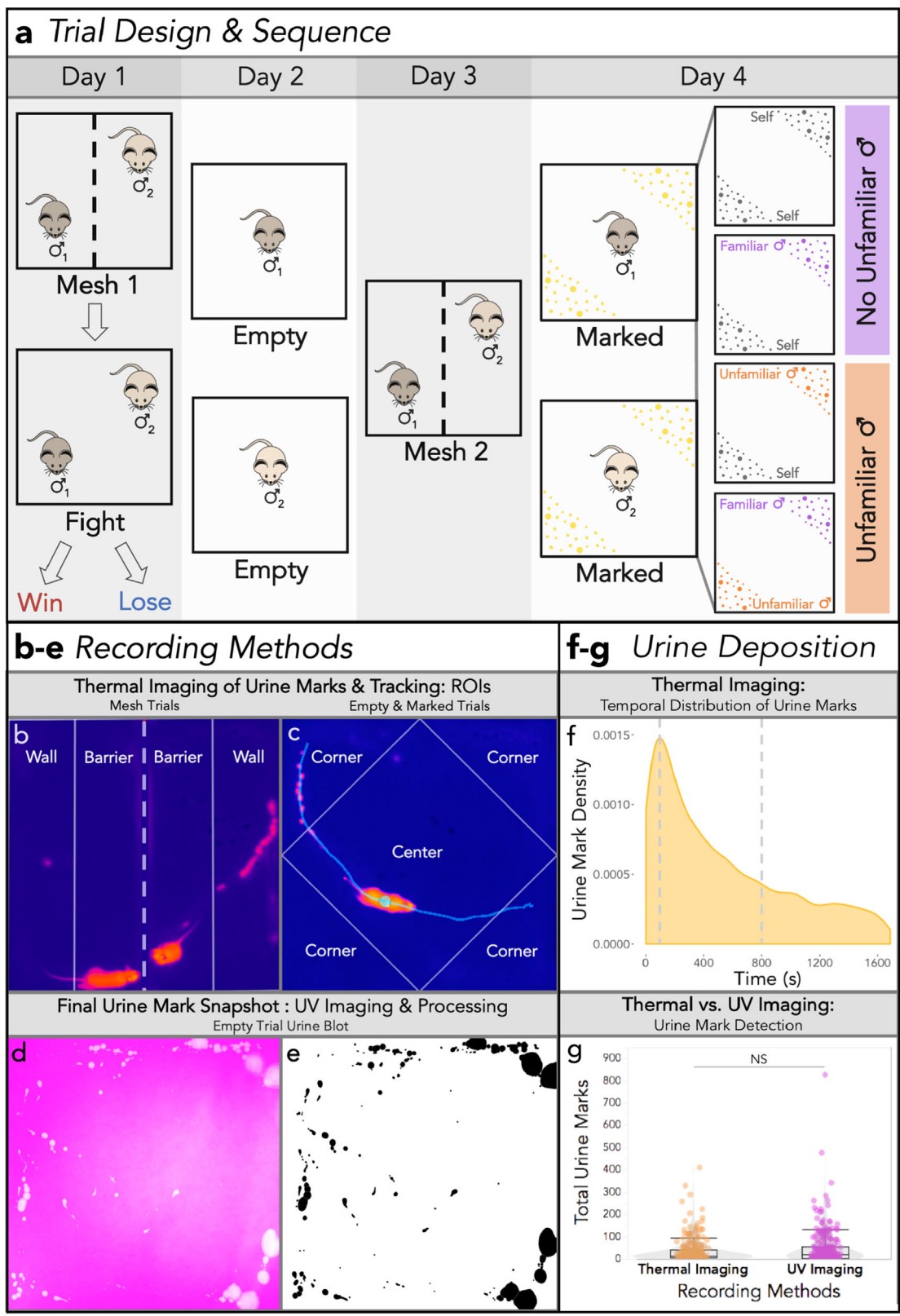

to-be losers, however, the two groups did not differ significantly (M2: $t_{1,112} = -0.69$, $p = 0.88$; Fig. 2a and Table S2). Post-fight, the total urine marks deposited by winners is significantly higher than losers (M2: $t_{1,112} = 30$ min, $p = 0.0001$; Fig. 2a and Table S2). Similar to previous studies[5,7,40–42], this relationship appears driven in part by a decrease in marking among losing males (M2: $t_{1,61} = 3.3$, $p = 0.006$), though post-fight winners

also trend towards being higher-marking (Fig. 2a, b and Table S2).

We next assessed the role of initial signal investment (# Mesh 1 marks) and fight outcome on subsequent allocation patterns (Fig. 2c). Given prior research, we expected some males would mark highly, lose the fight, and then suppress their marking[5]. Instead, we found that how much an individual marked pre-fight

**Fig. 1 Experimental design and recording methods. a** Trial design. Day 1: males were paired as competitors and placed into an arena separated by a mesh barrier indicated by a dashed line (Mesh 1). The mesh barrier was removed and males entered into an aggressive contest (Fight) concluding in winning or losing males. Day 2: each male was placed into an empty arena (Empty). Day 3: males were placed back into the mesh arena with the same (familiar) male competitor from the first trial (Mesh 2). Day 4: each male was exposed to one of 4 possible treatments of aliquoted male urine of 3 possible identities (self, familiar and unfamiliar) into two urine-marked zones. The 4 treatment groups: self-self, self-familiar male, self-unfamiliar male, familiar male-unfamiliar male. The familiar male stimulus is the urine of a male's paired competitor. **b** A thermal snapshot of a Mesh trial and the regions of interest (ROIs: Wall vs. Barrier) used to score urine mark depositions and track space use. The dashed line indicates the mesh barrier separating the two males. The solid lines depict ROIs males can traverse through on their side of the barrier. Urine marks are hot (orange-pink: close to the body temperature) on a cool (dark blue) ambient substrate (filter paper) temperature. **c** A thermal snapshot of an Empty trial (Day 2) with the ROIs used for scoring (Corners vs. Center) indicated with solid lines. The same ROIs were to score Empty (Day 2) and Marked (Day 4) trials. An example track of the mouse's trajectory two seconds before and after its current location is shown (light turquoise). **d, e** An example urine blot of an Empty trial imaged under UV light (D), and the processed inverted urine blot image (E: black spots: urine marks). **f** Density plot depicting the temporal distribution of all thermally detected urine marks across all trials. **g** Box and violin plot of the total number of urine marks detected across trials using thermal imaging and UV blot imaging recording methods (boxplot midline: median, box limits: upper and lower quartiles, whiskers: 1.5× interquartile range, points: outliers). A linear mixed model was used to model the relationship between recording method and the total urine marks detected (M1: Table S1). An analysis of variance was used to test for the overall effect of recording method (significance code: NS $p > 0.05$).

has a strong effect on the urine mark allocation post-fight (M3: $F_{1,59} = 9.2$, $p = 0.004$; Fig. 2c and Table S2). In other words, if you start off a low-marking individual you remain relatively low-marking, regardless of the fight outcome. Accordingly, both high-marking losers and low-marking winners are observed (e.g., Pair 3 in Fig. 2b). The pronounced winner-loser effects on urine allocation are therefore strongly modulated by initial signal investment.

Losing has a notable effect not only on the number of marks, but also where individuals place those marks in the arena (Fig. S3). To examine the spatial placement of urine marks, we split each side of the arena into two regions of interest (ROIs): (1) non-social wall and (2) social mesh barrier (Fig. S3a). In post-fight mesh trials, losers allocate their marks differently in space (at the wall vs. the barrier) depending on whether they started off as high or low-marking (Fig. S3a,b), suggesting losers may alter signaling strategies in addition to signaling effort. No such spatial allocation differences are observed among winning males (Fig. S3). We also examined space use patterns and found no differences (Fig. S3c). All individuals spend more time in the social region of the arena (barrier) regardless of fight outcome (Fig. S3c). Surprisingly, where males spend time does not correlate with where they mark (Fig. S3d), indicating males are not depositing urine where they spend the most time but are specifically allocating their urine marks in space.

**Social experience influences the temporal dynamics of scent-mark allocation.** In addition to the total number of urine marks, mice may alter the relative timing of urine mark deposition, such that marks are either more clustered or more evenly distributed in time. The relative timing of urine deposition provides novel information on the instantaneous rates of signaling, and reveals how mice choose to spend their urine reserves. A slow and regular mark deposition strategy is distinct from marking in rapid bursts.

We inspected the distribution of urine deposition events for winners and losers across mesh trials (Fig. 2d). Pre-fight, winners and losers display an initial peak at ~100 s (Fig. 2d). Post-fight the effects of fight outcome are clear, with winners marking more and losers less (Fig. 2d). The density curves, however, reveal that both winners and losers allocate more of their marks earlier in the trial post-fight (Fig. 2d). The shift to mark more rapidly regardless of fight outcome suggests a general priming effect of social competition on the timing of urine marking. This is evidenced by a diminished statistical difference between the winner and loser urine deposition distributions in the post-fight mesh trial compared to the pre-fight trial (Fig. S4b).

How quickly males place their first scent mark in the arena is strongly influenced by how highly or lowly they marked initially (M4: $F_{1,59} = 37$, $p = 9e{-}08$; Table S2). Similarly, trial order has a clear effect on the latency to mark (M4: $F_{1,58} = 10$, $p = 0.002$; Table S2), while fight outcome does not (M4: $F_{1,57} = 0.26$ $p = 0.61$; Table S2). The three-way interaction between trial order, fight outcome, and initial mark investment significantly effects mark latency (M4: $F_{1,58} = 12$, $p = 0.001$; Fig. 2e and Table S2). For both winners and losers, low-marking males are slower to mark than high-marking males, characterizing a low and slow pattern on the first day. Conversely, high-marking individuals mark rapidly upon entering the arena on the first day, representing a high and fast pattern. Across the two trials, winners mark more quickly, though this effect is scaled to their initial mark investment (Fig. 2e). Pre-fight, the initial peak in marking activity observed among losers (Fig. 2d) is primarily due to high-marking males (Fig. S4c). Post-fight, initially high-marking losers are slower to mark after losing, whereas individuals who initially marked infrequently speed up (Fig. 2e). After losing a contest males experience an equalizing effect on mark latency (Fig. 2e), suggesting there may be important priming effects of social competition on temporal marking features for both winning and losing males. Together this data demonstrates that complex changes in signaling behaviors are dependent on the initial signaling state of individuals.

We next examined the temporal rhythm of urine marking across mesh trials, which revealed unanticipated patterns. The intervals between urine deposition events differ noticeably pre- and post-fight, particularly when marks are made in close sequence to each other (Fig. 3a). Pre-fight, mark sequences have longer pauses between deposition events within mark series (Figs. 3a, S5). Whereas post-fight mark sequences are compressed, such that the time between marking events is shorter (Fig. 3a). To examine this relationship further we inspected the distribution of the inter-mark intervals (IMIs, i.e., the time between marking events) among winners and losers for both trials (Fig. 3b). Pre-fight, the most frequent IMIs are less than 3 s for winners and losers, though winners have a lower median mark value. (Figs. 3b, S5). Post-fight, there is a clear peak IMI of less than 1 s for both winners and losers (Fig. 3b). The overall median IMI is unchanged for winners but increases notably in losers, which is seemingly driven by the overall decrease in marking by losers.

To explore this shift in temporal dynamics within urine mark sequences, we classified sequential marking events that occur within 3 s as marking bouts (Fig. S5). Bouts can thus consist of a single mark or a series of marks (range: 1–27 marks). We then

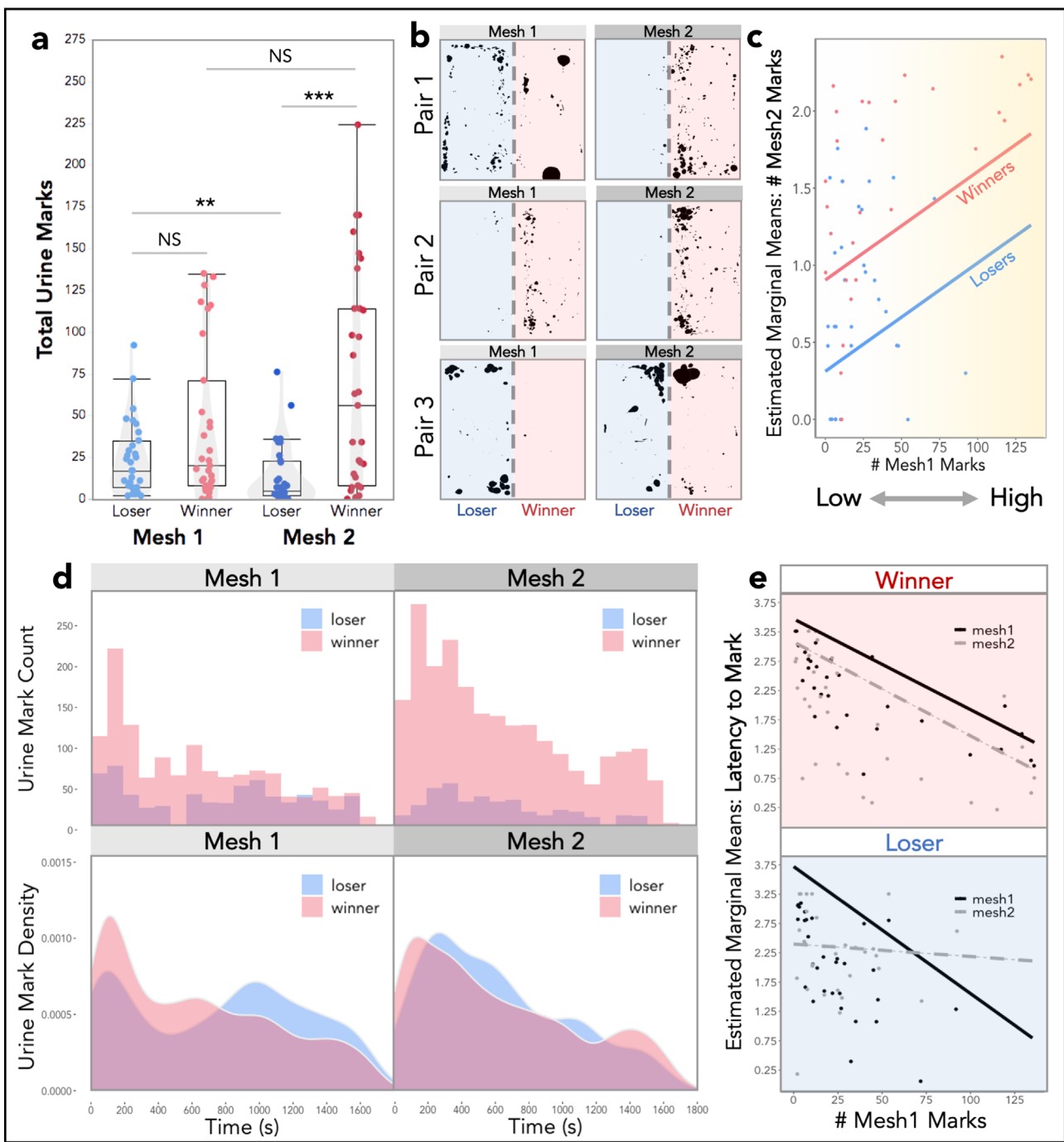

**Fig. 2 Male urine mark allocation in response to social competition across mesh trials. a** Total urine marks deposited in Mesh 1 (pre-fight) and Mesh 2 (post-fight) by losers and winners (boxplot midline: median, box limits: upper and lower quartiles, whiskers: 1.5x interquartile range, points: outliers). **b** Example mesh trial urine blots of three paired male competitors (winner and loser) pre- and post-fight. **c** Estimated marginal means of the total number of Mesh 2 marks (log-transformed) given fight outcome (winner: red, loser: blue) and initial signal investment (# Mesh 1 marks). **d** Histograms (top) of the temporal distribution of urine marks deposited by winners and losers in Mesh 1 (pre-fight) and Mesh 2 (post-fight) trials. Density plots (bottom) depict the density of urine mark deposition events over both 30-min mesh trials, distinguished by fight outcome. **e** Estimated marginal means of mark latency (log-transformed) in both mesh trials given the fight outcome and initial signal investment (# Mesh 1 marks). **a, c, e** Linear mixed models were used to model relationships (M2–M4: Table S2), analyses of variance were used to test for overall effects, and post hoc pairwise comparisons were performed using the emmeans package (significance codes: NS $p > 0.05$; **$p < 0.01$; ***$p < 0.001$). Dependent variables were logarithmically transformed to meet assumptions for model residuals.

examined the variation in IMIs within urine mark bouts (i.e., IMIs for bouts with 2+ marks, Fig. 3c). Trial has a strong effect on within-bout IMIs (M5: $F_{1,428} = 304$, $p = 2.0e-16$; Table S3), while fight outcome does not (M5: $F_{1,46} = 0.079$, $p = 0.78$; Fig. 3c and Table S3). Thus, marking events within bouts are more rapid

post-fight for winners and losers, indicating that competitive experience primes marking motor patterns, regardless of fight outcome. What is particularly striking, is that the observed shift from temporally extended mark "chains" pre-fight to temporally condensed mark "bursts" post-fight occurred after a single

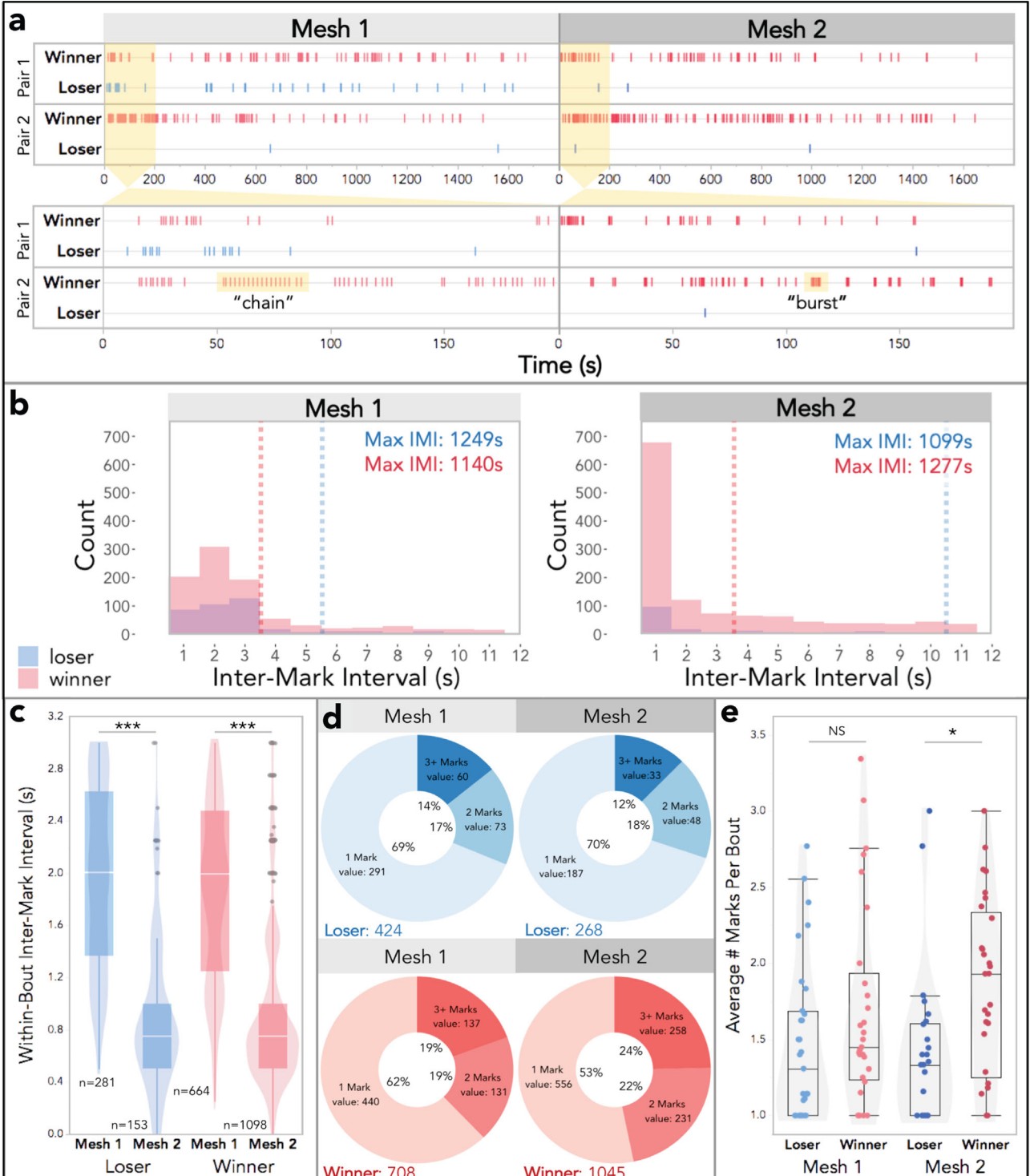

**Fig. 3 Temporal dynamics of urine mark allocation across mesh trials. a** Example event plots depicting urine marking of two pairs of male competitors over the course of both mesh trials for the trial duration (top) and a zoomed-in view of the first 200 s (bottom). "Chain"-like and "burst"-like marking bout examples are highlighted in yellow. **b** Histograms of the inter-mark intervals (IMIs) for winners and losers in both mesh trials. Median values are indicated with dashed lines. The range of IMIs extends to nearly the full trial length (only the first 12 s is shown). The maximum values are reported in the top right corner. Mesh 1: 65% of all IMIs are shown (<12 s), 57% of loser IMIs and 69% of winner IMIs. Mesh 2: 68% of all IMIs are shown (<12 s), 51% of loser IMIs and 72% of winner IMIs. **c** Box and violin plots of within-bout IMIs by fight outcome and mesh trial. **d** Donut plots by fight outcome and mesh trial depicting the proportions of bouts composed of: 1 mark, 2 marks, or 3+ marks. Mark totals are indicated (bottom left). **e** Boxplot of the average number of marks per bout by fight outcome and mesh trial. **c, e** Boxplot midline: median, box limits: upper and lower quartiles, whiskers: 1.5× interquartile range, points: outliers. **d, e** Linear mixed models were used to model relationships (M5-M6: Table S3), analyses of variance were used to test for overall effects, and post hoc pairwise comparisons were performed using the emmeans package (significance codes: NS $p > 0.05$; *$p < 0.05$; **$p < 0.01$; ***$p < 0.001$). Dependent variables were logarithmically transformed to meet assumptions for model residuals.

competitive encounter. We further investigated whether marking bouts are composed of 1 mark, 2 marks, or 3+ marks (Fig. 3d). Pre-fight losers have more single-mark bouts and winners have more multi-mark bouts (Fig. 3d). This relationship becomes even more stark post-fight. Losers decrease the overall number of marks across mesh trials, but the bout composition remains similar (Fig. 3d). Winners, on the other hand, increase the number of marks and alter their bout composition to include more multi-mark bouts (Fig. 3d). We compared the average number of marks per bout by fight outcome and trial (Fig. 3e). Bout composition is strongly affected by fight outcome (M6: $F_{1,58} = 10$, $p = 0.002$; Fig. 3e and Table S3). Post-fight, winners have a significantly higher average number of marks per bout than losers (M6: $t_{1,111} = -3.0$, $p = 0.01$; Fig. 3e and Table S3). This dataset reveals striking patterns of signaling behavior in male house mice that would otherwise have gone undetected without the use of thermal imaging.

**Dominance and familiarity interact to shape countermarking dynamics.** Given that males dynamically adjust marking behavior in response to social competition, we next explored allocation decisions toward the scent marks of other males. We were especially interested in whether males use knowledge of a recent competitor's identity in their signaling decisions, as males will competitively counter-mark to (i.e., mark over) the urine marks of other males[9,13]. While it is well-established that males alter marking behavior in response to fight outcome[5,7,8] and can finely discriminate urine identities[12,13], we have a limited understanding of how males implement this information in a competitive marking context. Do males adjust their scent marking behavior depending on their relationship to a male competitor? What role does familiarity play in signal allocation dynamics? We hypothesized that fight outcome would shape urine marking, and that familiarity would strongly govern signal allocation decisions.

To address these questions, we compared two trial types within the trial series in which no conspecifics were present: empty arena trials and urine-marked trials (Fig. 1a). The "Empty" trials contained no stimuli, and the "Marked" trials each contained two spatially distinct urine-marked zones of specific identities: their own urine (self: S), familiar male (FM) competitor urine, and/or unfamiliar male (UM) urine (Fig. 4b). The "familiar" males were the individuals each focal male was paired with during the mesh and fight trials for a total of 1.5 h, with whom they have an established dominance relationship (Fig. 1a). Familiar male (FM) urine was collected from this paired male competitor, who had a distinct genotype and major urinary protein profile from the focal male. Unfamiliar male (UM) urine was collected from novel adult males with a third genotype, and thus produced distinct major urinary protein profiles from either of the paired males. Importantly, this approach allowed all UM stimuli to be the same across subjects. Urine was collected one week prior to the start of the experiment. We examined responses to an empty arena and to the four different urine stimulus sets: S-S, S-FM, S-UM and FM-UM (Fig. 1a) by fight outcome and initial signal investment. Trial type (M7: $F_{4,76} = 5.2$, $p = 0.0009$), fight outcome (M7: $F_{1,83} = 27$, $p = 2e-06$), and initial signal investment (M7: $F_{1,58} = 32$, $p = 4e-07$), all significantly affect the marking behavior of males (Fig. 4a and Table S4). As does the two-way interaction between trial type and fight outcome (M7: $F_{4,77} = 5.8$, $p = 0.0004$; Table S4). Winners tend to mark more, and losers mark relatively lowly across treatment types. This pattern is observed in the responses to an empty arena (M7: $t_{1,100} = -3.9$, $p = 0.003$; Fig. 4a and Table S4). Notably, winners and losers show opposite responses toward familiar versus unfamiliar urine. Treatments without unfamiliar urine (Fig. 4a, b, purple: S-S and

S-FM) exhibit comparable marking responses in winners and losers (Fig. 4a). While it's perhaps less surprising that winners and losers mark comparably lowly to their own urine (S-S; M7: $t_{1,99} = -0.83$, $p = 1.0$), it is striking that winners and losers do not differ in their response to the S-FM treatment (M7: $t_{1,105} = -0.44$, $p = 1.0$; Fig. 4a and Table S4). Particularly for winners, as these males are not marking highly to the presence of another male's urine in the environment. The opposite pattern is observed in the presence of unfamiliar urine. Winners mark significantly more than losers to S-UM (M7: $t_{1,108} = -3.6$, $p = 0.009$) and FM-UM (M7: $t_{1,109} = -6.0$, $p < 0.0001$) treatments (Fig. 4a and Table S4).

We originally anticipated that in trials with two different urine identities males would differentially allocate urine towards each marked corner, we did not however detect any differences (Fig. S6a). It became clear while scoring trials that the space was too small to delineate marking to one stimulus corner versus the other, as males frequently deposit scent marks as they traveled through multiple regions of the arena. Our results also suggest that at this scale males mark in response to the most 'extreme' social odor in the environment (Figs. 4, 5). As a result, we consider each urine-marked treatment (Fig. 4b) as an entire scent environment, rather than as discrete subregions. Though we did not detect spatial differences in signaling within the spatial scale of these trials, we did detect region-specific differences in space use (Fig. S6b). Losers spend less time in the center ROI compared to winners ($t_{1,230} = -3.7$, $p = 0.007$), and spend less time in UM-marked corners relative to empty ones ($t_{1,199} = -3.5$, $p = 0.001$) (Figs. 1c, S6b).

Given that we observed very similar responses in the two treatments with unfamiliar urine present (S-UM and FM-UM) as well as the two treatments with only familiar urine (S-S and S-FM), we collapsed these similar treatments (purple: familiar-only male, orange: unfamiliar male) to further explore the role of familiarity and fight outcome on signal allocation (Fig. 4b–e). We standardized the marking behavior of males by calculating the difference in marks made in the empty arena trial relative to a scent-marked environment (Fig. 4c). The interaction between fight outcome and familiarity strongly shapes marking behavior in scent-marked contexts (M8: $F_{1,58} = 13$, $p = 0.0005$; Fig. 4c and Table S4). Winners increase the number of marks significantly more than losers in trials when unfamiliar urine is present (M8: $t_{1,58} = -3.0$, $p = 0.007$), whereas winners and losers do not differ when familiar-only scent marks are present (M8: $t_{1,58} = 2.0$, $p = 0.17$; Fig. 4c and Table S4).

We therefore find an inverse response among winners and losers toward familiarity (Fig. 4c). Winners mark highly to unfamiliar urine and lowly to familiar-only urine (M8: $t_{1,58} = -3.0$, $p = 0.01$), while losers mark lowly to unfamiliar urine and more to familiar-only urine (M8: $t_{1,58} = 2.2$, $p = 0.12$; Fig. 4c and Table S4). Notably, losers in the familiar-only treatment ($t_{1,14} = 4.5$, $p = 0.0005$) and winners in the unfamiliar treatments ($t_{1,16} = 4.4$, $p = 0.0004$) deviate significantly from zero, while their opposing treatments do not (shown in green: Fig. 4c).

**Temporal variation in signal allocation during countermarking.** The timing of signal allocation in scent-marked environments was also examined (Fig. 4d). In trials with no urine stimulus (Empty), winners allocate marks early in the trial (peak density ~100 s), while losers mark less with a later peak at ~250 s (Fig. 4d). In contrast, though the distributions are statistically distinct, winners and losers have quite similar density curves in familiar-only trials in terms of the timing of the initial peak (purple: S-S and S-FM) (Figs. 4d, S6d). What is also striking, is

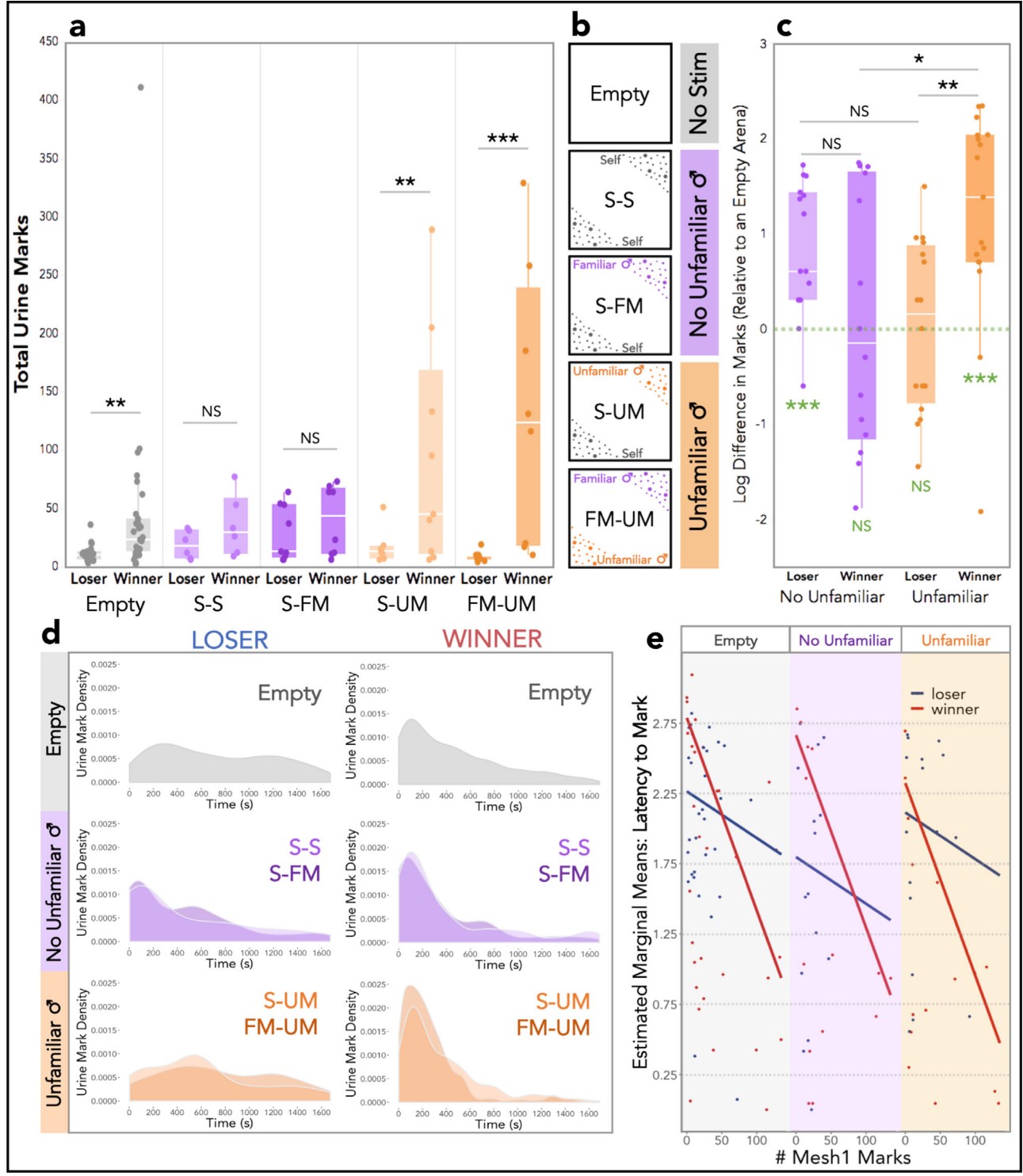

that the distributions of the S-S and S-FM trials are overlapping for losing males (D = 0.23, p = 1.0), and only moderately different for winning males (D = 0.13, p = 0.04; Fig. S6d). Though it's a slight effect, losers have a second late peak in marking activity in response to the S-FM treatment that is not observed to the S-S treatment (Fig. 4d). In trials with unfamiliar urine, winners and losers differ dramatically (D = 0.48, p = 2e−16; Fig. S6d). Winners quickly deposit large amounts of urine, creating a large initial spike in the density curves in S-UM (light orange) and FM-UM (dark orange) treatments (Fig. 4d). Losing males drop off and slow down their urine mark deposition, generating density curves with small

and delayed peaks (Fig. 4d). The temporal distribution of urine marks is therefore modulated by fight outcome and familiarity in scent-marked environments.

As the temporal dynamics of scent-marks were overlapping in trials either with or without unfamiliar male urine, we collapsed these into treatment groups (Fig. 4e). We further modeled the effects of treatment group, fight outcome, and initial signal investment, on the latency to mark (Fig. 4e). Mark latency is significantly predicted by the number of marks made in the first mesh trial, i.e., the initial investment recorded 3 days earlier (M9: $F_{1,57} = 10$, $p = 0.002$; Fig. 4e and Table S4). For winners and

**Fig. 4 Urine mark allocation across scent-marked contexts. a** Total urine marks deposited by winning and losing males in an empty arena and the four urine-marked treatments: self-self (S-S), self-familiar male (S-FM), self-unfamiliar male (S-UM) and familiar male-unfamiliar male (FM-UM). All males experienced an empty stimulus-free arena. Each male also experienced one of the four urine-marked treatments. **b** Schematic of the urine stimulus components for the empty and urine-marked treatments. Empty trials have "no stimulus" (gray), S-S and S-FM have "no unfamiliar male" urine present (purple), and S-UM and FM-UM trials have "unfamiliar male" urine present (orange). **c** The difference in total marks deposited by males in the urine-marked trials relative to the empty trials (log-transformed). Urine-marked treatments are grouped as "no unfamiliar male" urine (purple: S-S and S-FM) and "unfamiliar male" urine (orange: S-UM and FM-UM). Post hoc pairwise comparison significance values are indicated at the top of boxplots. One-sample t-tests (deviation from 0) significance values are indicated on the bottom of the boxplots (green). **d** Urine mark density plots of losing and winning males toward an empty arena, and to trials with no unfamiliar male urine: S-S (light purple) and S-FM (dark purple), and to trials with unfamiliar male urine: S-UM (light orange) and FM-UM (dark orange). **e** Estimated marginal means plot of mark latency in the empty trials (gray), urine-marked trials with "No Unfamiliar" male urine (purple), and urine-marked trials with "Unfamiliar" male urine present (orange), given the fight outcome and initial signal investment (# Mesh1 marks). **a, c** Boxplot midline: median, box limits: upper and lower quartiles, whiskers: 1.5× interquartile range, points: outliers. **a, c, e** Linear mixed models were used to model relationships (M7-M9: Table S4), analyses of variance were used to test for overall effects, and post hoc pairwise comparisons were performed using the *emmeans* package (significance codes: NS $p > 0.05$; *$p < 0.05$; **$p < 0.01$; ***$p < 0.001$). Dependent variables were logarithmically transformed to meet assumptions for model residuals.

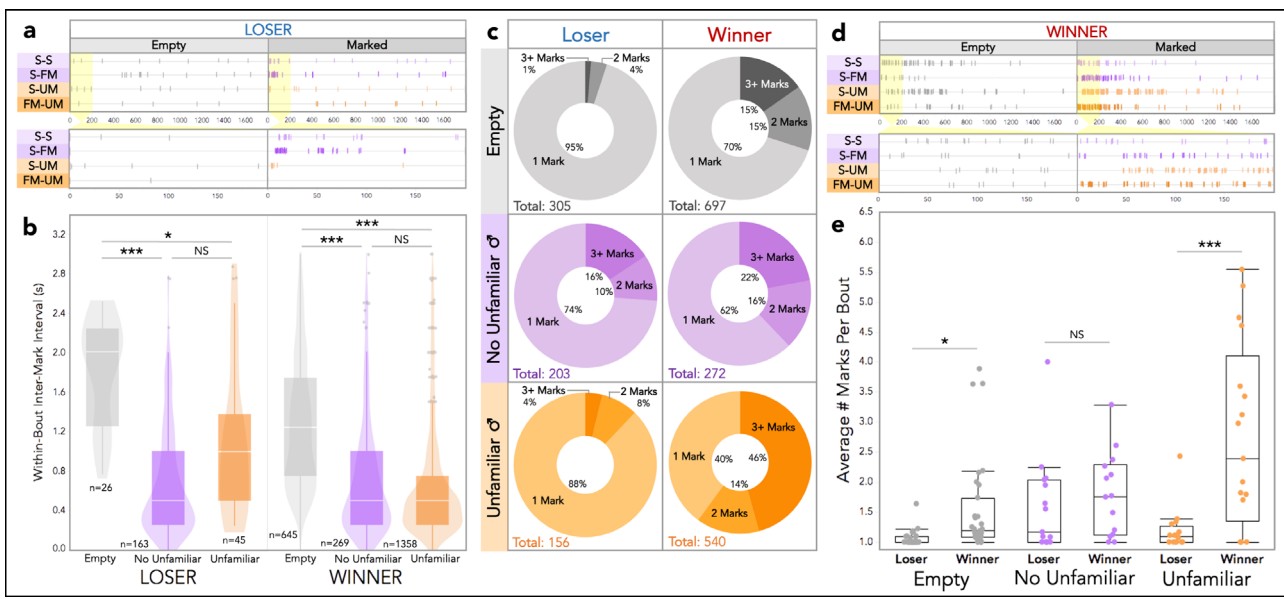

**Fig. 5 Temporal dynamics of urine signal allocation across scent-marked contexts. a** Example event plots depicting urine marking in Empty and Marked trials of four losing males, each exposed to one of the four different urine-marked treatments: self-self (S-S), self-familiar male (S-FM), self-unfamiliar male (S-UM), and familiar male-unfamiliar male (FM-UM). The event plot for the entire trial duration is shown on top and a zoomed-in view of the first 200 s is shown below. **b, c, d** Urine-marked treatments are grouped as "no unfamiliar male" urine (purple: S-S and S-FM) and "unfamiliar male" urine (orange: S-UM and FM-UM). **b** Box and violin plots of within-bout IMIs by fight outcome and trial group: Empty, No Unfamiliar (S-S & S-FM), and Unfamiliar (S-UM and FM-UM). **c** Donut plots by trial group and fight outcome depicting the proportion of bouts composed of: 1 mark, 2 marks or 3+ marks. Mark totals are indicated in the bottom left-hand corner. **d** Example event plots depicting urine marking in Empty and Marked trials of four winning males, each exposed to one of the four different scent-marked treatments: S-S, S-FM, S-UM, FM-UM. The event plot for the entire trial duration is shown on top and a zoomed-in view of the first 200 s is shown below. **e** Boxplot of the average number of marks per bout by fight outcome and trial group. **b, e** Boxplot midline: median, box limits: upper and lower quartiles, whiskers: 1.5x interquartile range, points: outliers. Linear mixed models were used to model relationships (M10-M11: Table S5), analyses of variance were used to test for overall effects, and post hoc pairwise comparisons were performed using the emmeans package (significance codes: NS $p > 0.05$; *$p < 0.05$; **$p < 0.01$; ***$p < 0.001$). Dependent variables were logarithmically transformed to meet assumptions for model residuals.

losers, initially low-marking individuals are slower to mark, and initially high-marking individuals are faster to mark (Fig. 4e). This relationship is most stark among winners, which exhibit steep slopes across treatment groups, while losers display more modest slopes (Fig. 4e). The interaction between fight outcome and initial signal investment, however, is moderate (M9: $F_{1,57} = 3.7$, $p = 0.06$; Fig. 4e and Table S4). The effect of fight outcome on mark latency is not significant (M9: $F_{1,56} = 1.3$, $p = 0.25$; Fig. 4e and Table S4). Treatment group on the other hand, significantly effects the speed of marking response (M9: $F_{1,75} = 3.2$, $p = 0.048$; Fig. 4e and Table S4). Losers mark most rapidly in familiar-only trials, and winners mark most rapidly in

trials with unfamiliar urine (Fig. 4e). The intersection points of the linear models for winners and losers reveal additional insights. Winners transition to a more rapid marking response relative to losers differently across treatment groups depending on initial signal investment. In familiar-only trials, only the initially very high-marking (>85 marks) winners mark more rapidly than losers, other winners are slower to mark. The opposite is true in trials with unfamiliar urine, in which even initially low-marking (>20 marks) winners mark more rapidly than losers (Fig. 4e). This demonstrates that initial signal investment significantly predicts aspects of marking behavior, including the temporal allocation of urine marks.

We next examined the timing and composition of marking bouts (Fig. 5). More chain-like bouts are observed in no-stimulus empty trials, whereas more rapid bursts of urine marking are produced in scent-marked trials (Figs. 5a, 5d, S6c). Therefore, over the 4-day trial series males mark increasingly in bursts, suggesting competitive experience shapes temporal features of signal allocation. To explore this further we looked at within-bout IMIs (Fig. 5b). Both fight outcome (M10: $F_{1,64} = 6.2$, $p = 0.02$) and treatment group (M10: $F_{1,152} = 40$, $p = 1e-14$) significantly effect within-bout IMIs, with a modest interaction (M10: $F_{1,154} = 2.5$, $p = 0.08$; Fig. 5b and Table S5). As expected, the within-bout IMIs are significantly longer in empty arena trials than either scent-marked treatment groups for winners or losers (Fig. 5b). Winners, however, marked with similar rapid bursts (short IMIs) regardless of familiarity with the urine stimulus (Fig. 5b). Conversely, losers tend to mark in bursts specifically during familiar-only trials (Fig. 5b). This bout timing is most prominent in the S-FM trials (Figs. 5a, S6c), which reveals losers distinctly adjust the temporal dynamics of their urine marks based on the identities in the environment. It is striking that, again, losers signal most conspicuously toward males who recently defeated them in a competitive contest.

The number of marks per bout changes with social outcome and scent-mark type (Fig. 5c). The average number of marks deposited per bout is significantly shaped by scent-mark familiarity (M11: $F_{1,76} = 13$, $p = 1e-05$) and fight outcome (M11: $F_{1,66} = 22$, $p = 2e-05$), with a strong two-way interaction (M11: $F_{1,76} = 6.3$, $p = 0.003$; Fig. 5e and Table S5). Fight outcome and familiarity both influence the composition of marking bouts (Fig. 5c, e). In an environment empty of scent marks, winners allocate considerably more multi-mark bouts than losers (30% vs. 5%; Fig. 5c), and the average number of marks per bout is significantly higher among winners (M11: $t_{1,105} = -3.0$, $p = 0.03$; Fig. 5e and Table S5). Interestingly, the differences in bout composition narrows in scent-marked trials with familiar-only urine (Fig. 5c). In these trials, winners deposit slightly more multi-mark bouts (38%), while losers dramatically shift the amount of multi-mark bouts (26%; Fig. 5c). The average number of marks per bout does not differ between winners and losers in familiar-only trials (M11: $t_{1,117} = -1.0$, $p = 0.87$; Fig. 5e and Table S5). The reverse is true for trials with unfamiliar male urine present (Fig. 5c). Here, losers produce bouts with similar bout compositions to the empty arena trials (Fig. 5c). Winners double the proportion of multi-mark bouts compared to empty arena trials (60%), and many bouts contain at least 3 marks (46%; Fig. 5c). The average number of marks per bout is significantly higher among winners when unfamiliar urine is present (M11: $t_{1,117} = -5.6$, $p = <0.0001$; Fig. 5e and Table S5). Thus, the temporal rhythm and composition of urine allocation patterns change in response to the presence of different urine identities in the environment.

## Discussion

Using a thermal imaging approach, we discovered context-dependent dynamic and static responses in urine mark allocation, latency, and rhythm, toward competition and variable social environments (Figs. 2, 3). Collectively, these data provide strong evidence that male mice remember their experiences with other individuals, and update their decisions based on this information. Winning or losing has strong and long-lasting effects on signaling decisions (Figs. 4, 5), most prominently on total allocation effort and marking bout composition. As described in the literature, we find males quickly downregulate urine allocation after losing a competitive contest[5,40,41]. However, we also find that initial signal investment has stable and robust effects on marking behavior. In

other words, where males start off predicts their signaling decisions days later. Low-marking individuals remain relatively low-marking, and high-marking individuals remain relatively high-marking. The magnitude of the observed winner-loser effects is therefore contingent on the initial investment decisions of males.

Our data demonstrate that male house mice dynamically adjust their signal allocation and timing depending on the social scent landscape. Prior studies have shown that male mice finely distinguish self from non-self urine[9,13], and that females recognize specific males based on their urine marks[12,24]. However, it has been less clear whether males use such information in territorial contexts. Here we find that signaling decisions are profoundly shaped by contest outcomes and familiarity with male competitors. Surprisingly, losers tend to increase mark allocation effort and display more frequent bursts of multi-mark bouts toward familiar male urine marks. In contrast, winners downregulate their marking efforts toward familiar urine. The responses toward familiar males are even more stark when compared to how males respond to unfamiliar male urine. A key component of our design was that a single standardized male urine stimulus was used across all trials as the unfamiliar male. Therefore, the differences in responses to unfamiliar male urine must be attributed to shifts in mouse behavior as a result of recent experience rather than due to variation in the social odors they were presented. Winners dramatically upregulate all competitive marking efforts and losers go scent "silent." Under the dear enemy model, territorial males should remain vigilant toward unfamiliar males, as they threaten their current dominance status[25–29,43]. In our study, winners echo these predictions of the dear enemy model by increasing signaling efforts toward novel male urine. Losers on the other hand are at risk of further aggression. By staying "silent" losers may avoid conflict with a new territorial contender, potentially in a "wait-and-see" strategy[44]. However, when presented with urine of the male that recently defeated them, losers actually upregulate marking efforts. This response may be a "nasty neighbor" effect, in which the threat of familiar territorial males exceeds that of strangers[27]. Alternatively, this increased marking response could be a form of subordinate marking. Importantly, the scent-marked experiments were not performed in males' home territories (i.e., home cages) and males did not maintain territory boundaries with each other. As a result, whether males perceived the arena as an extension of their home territory, as a shared home range, or as a disputed territory is unclear. Future work investigating scent intrusions on home territories will be necessary to more directly elucidate the extent of the dear enemy and nasty neighbor effects in house mice. Nevertheless, we find that recent social experience and familiarity modulates how much animals invest in territorial advertisement and signaling.

Competitive experience also has strong priming effects on temporal features of scent marking. Mice mark more rapidly after a contest, regardless of outcome (Fig. 2). Similarly, the time between deposition events shrinks, such that marking bouts transition from prolonged chain-like sequences to rapid bursts. Aggressive contests likely shift males into a competitive state, driving changes in urinary motor patterns. Surprisingly this occurs after just a single aggressive contest. Voluntary, involuntary, and context-dependent urination are all mediated by neuronal subpopulations in the Barrington's nucleus in the brainstem[30,32,33]. The fine-scale adjustments in urinary motor control we observe reveals additional complexity to this underlying circuitry, opening avenues for future research to examine how competitive interactions and social signals modulate motor outputs.

We unexpectedly detected a cohort of "silent" low-marking winners, for which we could find no prior description of in the literature, suggesting several possible hypotheses. First, the result

may be driven in part by our trial design. By pairing evenly-matched males, we may have observed more instances of low-marking males winning. Furthermore, better-than-expected outcomes could give rise to slower response times than worse-than-expected outcomes, in which high-marking losers rapidly downregulate signaling effort[5,40,41]. Second, low-marking males may differ in some aspect of hydration physiology. Species and strains of mice vary in water intake and urination levels[45–47], though we observed low-marking winners in both strains used in this study. Third, "silent" competitive males might represent a distinct signaling strategy in house mice. Given the high metabolic costs of signaling, it's plausible that some males might withhold signal investment to continue investing in body mass or to avoid detection by other males. Male house mice therefore may pursue diverse signaling strategies, including the classically described "territorial males" that invest highly in urine marking as well as scent-silent "sneaker males"[48–50]. While our data does not directly test this relationship, the frequency of low-marking winners warrants further investigation. Certainly, the simple correlation between marking and dominance is considerably more complex than previously described.

This work emphasizes the importance of examining signaling behaviors across social contexts in order to examine the decision rules for costly and complex behaviors. Furthermore, the implementation of thermal recording in social behavior assays has the potential to reveal important features underlying the neurophysiological basis of socially-modulated and voluntary urination behaviors.

## Methods

**Experimental animals**. All experimental subjects in this study were males ($n = 62$) from two wild-derived inbred strains (NY2 and NY3) of house mice (*Mus musculus domesticus*). The progenitors of these strains were captured near Saratoga Springs, NY in 2013 by MJS[51] and are related to the SarA/NachJ, SarB/NachJ and SarC/NachJ strains now available from the Jackson Lab. All collection sites were at least 500 meters apart to avoid collecting from closely related mice[51]. Wild-caught animals were mated, and sibling-sibling pairings were performed since August 2013 to generate inbred lines[51]. While from the same general mouse population, the lines used here are not closely related and are expected to differ by millions of single nucleotide polymorphisms[51,52]. At the time of experimentation these lines had experienced roughly 12 generations of inbreeding. Wild-derived strains were used because naturalistic competitive behaviors are less pronounced in highly inbred and domesticated laboratory strains[53,54] and inbred strains tend to share identical urinary protein profiles[55]. Individual wild house mice have distinct blends of urinary proteins that are used to each recognize other[12,56,57]. We therefore wanted to ensure that all interacting males smelled distinct in an ecologically relevant manner. In other words, we wanted to ensure that experimental mice assessed and interacted with genotypically distinct individuals. At weaning age (3–4 weeks) males were placed into a holding cage alone for 1–2 weeks, and were subsequently paired with a female to allow for sexual experience, as sexually naïve mice are known to exhibit different social behaviors[58]. All males were allowed to reach adulthood (3–5 months old by the time of experimental testing) and had the opportunity to produce one or more litters. All cages contained corn cob bedding, cardboard huts, and cotton nestlets. Mice were maintained in an Animal Care facility at Cornell University with a 14:10 shifted light:dark cycle (dark period: 12 p.m.–10 p.m.) and were provided food and water ad libitum. Mice were handled minimally and with transfer cups whenever possible to reduce stressful handling. All experimental protocols conducted at Cornell University were approved by the Institutional Animal Care and Use Committee (IACUC: Protocol #2015-0060) and were in compliance with the NIH Guide for Care and Use of Animals.

**Behavioral experiments**. One day prior to experimentation, we recorded subject male body weights to size-match individuals as closely as possible (average weight difference: 2.4 g). All males were in breeding cages at the time of the experiment and most successfully reproduced (84%) prior to start of the trial series. As house mice are nocturnal, all experiments were conducted in the dark during the dark cycle[59]. All experimentation occurred between 12 p.m.–5 p.m. to minimize daily circadian variation. Trials were performed between the months of May–November during the years of 2018 and 2019. The winter months were avoided to prevent seasonal affects in which the mice are less active. Laboratory mice exhibit seasonal variation with respect to certain physiological parameters like serum concentrations of sex hormones, suggesting a possible mechanism for an internalization of annual time independent of light cycle, temperature and humidity[60]. While the

available literature provides conflicting evidence as to whether these effects extend to behavior, we nonetheless took measures to avoid such confounds[61], particularly as the mice were wild-derived and recently inbred from the northeastern United States. Trial series were performed in sets of 2–5 male pairs.

Behavioral trials consisted of a 4-day trial design, in which age and weight-matched adult breeding males of distinct wild-derived strains (NY2 and NY3) were paired as competitors and presented a series of social and scent-marked trials (Fig. 1a). We pair-matched each NY2 mouse with a NY3 mouse to ensure that no two paired mice were genotypically identical and that their scent marks were perceptibly different (unique major urinary protein profiles)[13,55–57], resulting in a total of 31 pairs ($n = 62$). All house mice within an inbred strain have identical major urinary proteins (as a result of inbreeding)[55–57]. Because major urinary proteins are used in recognizing individuals[9,12,13,56], we wanted to ensure paired males had distinct urine profiles. Prior to experimentation males only had exposure to their own strain and thus their own MUP type, all other MUP profiles would be novel.

To guarantee identification of males within a pair (NY2 and NY3 strains are visibly indistinguishable), we ear-clipped and bleached a patch of rump fur of one male in each pair a week prior to experimentation. Mice were anesthetized with isoflurane (5%). A heating pad was used to maintain a stable body temperature. Isoflurane was delivered at 1–3% throughout the bleaching procedure. L'Oreal Colo Rista Bleach Ombre (salon bleach) was mixed as per the manufacturer's instructions and dabbed onto the top layer of fur using a sterile cotton swab. Care was taken to prevent bleach from contacting the skin. Twenty minutes after application, sterile cotton tipped swabs dipped in water were used to rinse the bleach from the fur. The fur was then dabbed dry with paper towels. Mice were placed under a heat lamp for 5 min or until they were fully recovered from anesthesia before being transferred back to their home cage.

All trials were performed in one of two trial chambers that were sound proofed and fitted with recording systems. For all trials large sheets of Whatman filter paper lined the floor of each trial to collect urine blots and to present urine stimuli. The same size PVC arenas were used throughout (50 cm × 50 cm), though split in half with the mesh barrier for the Mesh trials (Fig. 1a). At the end of each trial, males were placed back into their breeding home cages. On Day 1 of the trial series, paired males were placed on either side of a wire mesh barrier in an arena for 30 min (Mesh 1, Fig. 1a). At the end of the 30 min, males were briefly removed from the arena into large transfer cups, the filter paper was labeled and removed, a fresh filter paper was placed in the arena, and the mesh barrier was removed. Males were placed back into the arena for a 30-min aggressive contest (Fight, Fig. 1a). A single extended contest was used to ascertain dominance between paired males because the arena was larger than a standard resident-intruder assay (18 × 28 cm) and males were well-matched, so we expected contest resolution would take longer[20]. We opted for a single contest as opposed to a contest series to minimize the physical toll of the fight trials and the stress of added handling. On Day 2, each male was placed alone in a stimulus-free empty arena for 30 min (Empty, Fig. 1a). On Day 3, males were placed back into the mesh arena for 30 min with the same male competitor encountered on the first day, without the subsequent fight trial (Mesh 2 trial, Fig. 1a). On Day 4, males were placed into the arena alone for a 30-min urine-marked stimulus trial, consisting of one of 4 possible treatment types. Each treatment included two spatially distinct urine-marked zones placed in opposite corners of the arena (front right—back left vs. back right—front left). Urine-marked corner zones contained aliquoted male urine of 3 possible identities: self, familiar, or unfamiliar male. Urine stimuli were placed on the filter paper directly before the trial start in standardized locations and volumes. The four treatment types span a range of scent-mark combinations: self-self, self-familiar, self-unfamiliar, familiar-unfamiliar. Paired males (winner-loser pairs) received the same urine-marked stimulus treatment, with the exception of three pairs due to urine collection constraints. For all trials (Days 1–4) the first and last minute of each trial was trimmed prior to analysis. This was done to minimize detection of startle-based urination events caused by placement of mice into arenas and any jostling caused during trial set-up and take-down. The total analyzed trial length was thus 28 min.

Trials and treatments were randomized as follows. Male trial order and arena chamber was pseudo-randomized each day to avoid confounds in arena location and marking behavior over the course of the designated trial period. The orientation within the Mesh 1 trials was also randomized (whether males were placed near the back or front of the arena) to account for variation in sound disturbances for males closer to the chamber door; orientations were subsequently flipped for each pair in Mesh 2. Urine-marked trial treatments were pseudo-randomly assigned to each male pair, to ensure similar numbers of male pairs were exposed to the 4 treatment types across sets of trials series. The orientation of urine stimuli was randomly assigned to corner orientations (front right—back left vs. back right—front left). Lastly, the fur bleaching for male identification was performed on one mouse strain (NY2 or NY3) for each trial set, but the bleached strain was switched between trial sets to prevent errors within a trial set and to avoid bleaching only one strain across trial sets.

**Urine collection**. Urine was collected from each experimental male subject to present self and familiar male (paired competitor) urine in the urine-marked zones on the final day of the trial series (Fig. 1a). For unfamiliar male urine, we collected

from a third distinct inbred mouse line (C57BL/6), to again ensure that the novel male urine presented had a distinct urinary protein profile from experimental individuals. This was necessary in order to have the urine scent marks of self, familiar male and unfamiliar male be distinguishable as different urine identities in the environment, given that mice have been shown to use major urinary proteins in this capacity[9,12,13,56]. Males cannot distinguish inbred within-strain individuals apart by their urine alone[16] as same-strain males share the exact same MUP profile and thus smell like "scent twins"[16].The use of three distinct genotypes also reflects natural ecological conditions, where individuals tend to interact and compete with multiple genetically distinct individuals. Inbreeding avoidance, cooperative nest mate recognition, and male countermarking all use a self-referential MUP-matching mechanism[12,13,62,63]. Even very slight differences in urine profile from a male's own urine, such as the addition of a single novel MUP or change of ratio among existing MUPs, is sufficient to elicit increased countermarking behavior[12,13]. Therefore, the exposure to the urine of a third distinct MUP profile would have been entirely novel. Regardless of how similar or different urine profiles are across the 3 strains, house mice would have no means to ascertain closer versus more distant relatedness outside of this self-matching mechanism (which only occurs between siblings)[12,13,62,63].

Urine collection was performed using the single animal method: males were placed atop a metal grate (an upside down cage hopper) over a clear plastic bag for 30 min to 1 h[64]. Males were subsequently taken off the plastic bag and returned to their breeding cage. The urine droplets present on the plastic bag were collected and stored at −80 °C until use. Urine collected from subject males was stored individually until the day of the urine-marked trials (Day 4: Fig. 1a). For sufficient urine volume for the urine-marked trial treatments (Fig. 1a), between 200 and 400 μL was collected from each NY2 and NY3 male subject. On the day of the urine-marked trials, individual aliquots for a subject male were thawed on ice and pooled together. For unfamiliar male urine we collected a large batch of urine from over 20 adult breeding C57BL/6 males. Urine was stored on the day of collection at −80 °C. Once a sufficient volume was collected to use as stimuli across all trials, individual aliquots were thawed on ice, and all C57BL/6 male urine was pooled into a single volume and subsequently aliquoted and stored at −80 °C. Pooling the urine of individuals of the same strain does not affect the MUP profile, as all inbred individuals share the same exact profile[16]. We pooled the C57BL/6 ("unfamiliar") male urine to ensure that the urine stimuli presented to males was exactly the same across all treatments and trials, without any individual-specific effects. However, while the MUP profiles do not vary across individuals, the overall levels of MUP production (high vs. low) do vary from male-to-male[14,65,66] as do various volatiles[67–69]. Therefore, by pooling C57BL/6 male urine, all experimental males were exposed to the exact same "unfamiliar male" urine stimulus, with the same MUP profile, the same MUP levels, and the same blend of volatiles.

The overall level of MUP production varies with the age, sex, physiological status, and social status of individuals[9,14,65,66,69]. We controlled for this by only collecting urine from males that were fully adult and were housed in breeding cages (i.e., maintained a territory and had sexual experience). Competition and dominance interactions have also been shown to affect MUP production, with more dominant individuals producing MUPs in greater quantitites[14,70]. It takes at least 2 weeks for this dominance-modulated shift in MUP production to occur, thus within the time frame of our experimental design (4 days) there would be insufficient time for protein levels to shift in the urine profiles of our experimental males.

**Recording methods.** All trials were recorded with a security camera system (iDVR-PRO CMS) at 1080p and 30 frames per second to visualize the high-speed aggressive encounters and to clearly distinguish the male identities (ear-marked and bleached fur). All trials (including fight trials) were recorded thermally using an infrared camera system (PI 640; Optris Infrared Sensing). Thermal cameras were fitted with 33° × 25° lenses and mounted above the experimental arena chambers such that field-of-view for each camera covered the entire arena. The thermal detection window was set at: 61 °F–107 °F. Data frames were collected at the max speed, averaging at 3 Hz. Thermal video data was saved by screen-capturing live Optris video output using OBS Studio software. Raw temperature data was also collected in semicolon-delimited CSVs, providing a readout of the temperature in each pixel for each frame.

**Behavioral scoring and analysis.** All videos were scored by a blind observer using Behavioral Observation Research Interactive Software (BORIS)[71]. For the fight trials (Fig. 1a), we scored the following aggressive behaviors: chasing, hitting, boxing, and wrestling bouts (Fig. S1c) using the infrared security camera video recordings. To score urine mark deposition events Optris thermal video recordings were used for all trials. Urine depositions were scored as a clear hot spot following the focal mouse's trajectory that subsequently cooled below substrate temperature. Urine marks placed in close spatial and temporal proximity were considered separate deposition events if at the moment of deposition there was a detectable cold barrier line separating the urine marks. Urine depositions are distinct from urine distribution events. Deposition events require the detection of a hot spot that subsequently cools. Distribution events are urine marks which start off cool. Mice will also create such "cool spots" by walking through a recently deposited urine mark, and tracking this cooled urine with their paws or tail. These small

distribution events are not counted as scent marks in this study. Instances of "overmarking" were also observed, i.e., where males place a urine mark directly on top of a previously existing mark. Overmarks were easy to distinguish with thermal recording as a hot urine spot was deposited on top of a prior and thus cooled urine mark. Notably, such overmarks would not be detectable using standard recording methods as the urine marks would bleed together as appear as a single mark if counted at the end of the trial. Fecal depositions could be eliminated as they are frequently cooler upon deposition than urine, cool much more slowly, have a distinct shape, and are typically moved around the arena quickly by the mice.

In addition to scoring the timing of urine deposition events, the placement of urine marks was also scored. Using screen annotation software, we drew precise lines on the video observation corresponding to regions of interest (ROIs) for each thermally-recorded trial (Fig. 1b, c). For the mesh trials, each side of the arena was split into two equal halves corresponding to either the social "barrier" region or the non-social "wall" region (Figs. 1b, S3). For the empty and urine-marked trials the arena was split into a total of 5 ROIs: 4 equal-sized corners and a center region. Each corner triangle-shaped ROI connected at the midpoint along each arena wall (Fig. 1c). In the urine-marked trials some corners contained urine stimuli and some were empty (Fig. 1c, S6a, b). In the scent-marked trials with two different urine identities (Fig. 1a), we had anticipated males would differentially allocate urine towards each marked corner. We did not detect clear effects of differential allocation to marked corners (Fig. S5a), suggesting two non-mutually exclusive possibilities: (1) the scent of unfamiliar males may be more important in driving allocation decisions and (2) the size of the arena may be too small for delineated corner-based allocation. While scoring the trials, it became clear that the space was likely too small, as males frequently walked through corners while performing a scent-mark bout that extends across multiple ROIs. As such, analyses focused on the presence of familiar and unfamiliar urine in the entire arena environment.

**Tracking.** Mice were tracked using the software UMATracker (Release 12)[72]. Infrared security camera recordings were used to track focal mouse movement, as these videos were recorded at a higher framerate. Filters were generated using the following modular settings (in order): output—Closing: Kernel = 6—Opening: Kernel = 6—Threshold: 100—BGRToGray—input. Videos were tracked using Group Tracker GMM algorithm. Area51 was used to generate desired regions of interest for each trial (Fig. 1b, c) and analyze the relative space use in each of these regions. For the mesh trials, each side of the arena was split into two equal halves corresponding to either the "barrier" or "wall" ROI (Figs. 1b, S3). For the empty and urine-marked trials the arena was split into a total of 5 ROIs: 4 equal-sized corners and a center region. Each corner triangle-shaped ROI connected at the midpoint along each arena wall (Fig. 1c). The midpoints along the edges of the arena were used to consistently generate the ROIs using the Area51 software, even when the angle of the camera relative to the arena differed by slight degrees. The R package *trajr*[73] was used to quantitatively characterize the following information from the tracked data frames: speed, acceleration, and trajectory length.

**Urine blot imaging and processing.** Trials were run on Whatman filter paper substrate. Arena edges were outlined with pencil on the filter paper at the end of each trial. We collected all sheets of filter paper used in experimentation (except for the Fight trial) and photographed them under ultraviolet (UV) light. We used three UV bulbs to evenly distribute light on the large filter paper area. Images were converted to greyscale in Adobe Photoshop and the magentas were reduced to ~20% to observe edges of urine marks clearly. Greyscale images were subsequently processed in ImageJ (Fiji). We subtracted background pixels for a cleaner image (100 px), applied image thresholding (manually adjusted when necessary), and converted images to binary in order to convert to mask, fill holes and perform watershed algorithm. This processed image was then used to analyze the number of particles, with Size (pixel^2): 100 − Infinity and Circularity (0–1.00).

**Urine mark bout classification.** The median inter-mark interval (2.99 s) for all males across all trials was used to determine whether marks get clustered into a marking "bout" (Fig. S5a). Any two marks that occur in sequence with an inter-mark interval less than 3 s are clustered together into a multi-mark bout, allowing us to examine within-bout dynamics. Other clustering methods were attempted but were less successful at classifying marking bouts, when visually checked with scored videos.

**Statistics and reproducibility.** We conducted all statistical analyses in R 3.6.0 (R Development Core Team 2019). We used linear mixed models (Tables S1–S5) and paired statistical tests to examine relationships between dependent. Models were fitted using the package *lme4*[74]. The *lmerTest* package was used to calculate degrees of freedom (Satterthwaite's method) and p-values[75]. Dependent variables were transformed for a subset of models to meet assumptions for model residuals after visually inspecting model residuals (all transformations are clearly indicated in model details in Tables S1–S5). We used a type 3 analysis of variance to test for overall effects of fixed factors or interactions in the models (Tables S1–S5). For all paired analyses male ID was included as a random effect. Post hoc comparisons were conducted using the *emmeans* package[76]. R script and datasheets used for all statistical analyses are provided in the Supplementary Material.

**Reporting summary**. Further information on research design is available in the Nature Portfolio Reporting Summary linked to this article.

## Data availability

All Supplementary Tables (Tables S1–S5) and Figures (Figs. S1–S6) are provided in the Supplementary Material. All datasheets used analyses, in addition to compiled raw and summary datasheets, are included in the Supplementary Data. A readme file provides details about the data organization and naming.

## Code availability

The annotated R code (stats.R) used in all analyses are included in the Supplementary Data (along with the corresponding datasheets).

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

## Acknowledgements

We thank Kevin Besler, Kusuma Anand, Christen Rivera-Erick and Melanie Colvin for crucial technical assistance; Russell Ligon and Caleb Vogt for helping establish recording systems and tracking methods in the lab; and James Tumulty for paper feedback.

## Author contributions

C.H.M. and M.J.S. conceived the study. C.H.M. performed trials and analyses. M.F.H., J.Y., B.C.C., K.H. and A.Y.L. collected samples, scored behavioral trials, and generated tracking data. C.H.M. wrote the initial drafts of the paper. C.H.M., M.R.W., and M.J.S. edited the paper. All authors contributed to paper preparation.

## Competing interests

The authors declare no competing interests.
