## [Peer Review File · Communications Biology]

Reviewers' comments:

Reviewer #1 (Remarks to the Author):

Revision

The article is very interesting, relevant, and adequate, but there are some issues we need to discuss and corrections to be made. Your article reaches several audiences by contextualizing the results with neuroscience data. But I have some doubts about the cognition/perception of mice that need to be clarified. The two strains selected are from the same location, this may have interfered with the recognition of familial and unfamiliar individuals. You could have the same results without the fight trial.

Ln 82 – 85 There was just one competitive context (fight trial) between the pairs of male mice, and that made then a familiar pair in your experiment. Why were the mice exposed to only one fight trial? What were the criteria for defining that pair of male mice are familiar to each other? As nothing about how animals are housed was mentioned in your article, it might be interesting to bring more details about this.

Ln 382 – 383 Why did you test only in two strains? Did the effect hold in a third strain of mice compared to these two (NY2 and NY3)? Are the urinary proteins profiles of NY2 and NY3 more similar with each other than are they similar with the C57BL/6 urinary protein profile? Two strains from the same site (Saratoga Springs) were selected to be familiar pair and a different strain was selected from another site to be an unfamiliar (C57BL/6), it is not possible to say whether the effect was due to genetic differences in recognition (differences in protein profiles) or due to a difference in experimental treatment. Mice could respond differently to more similar and more different genetic profiles (urinary protein profile), and your experiment did not allow for that distinction.

Ln 343 – 351 Your conclusion did not seem to consider that the disputed territory had a winner, and that winner could be the owner of the territory. As the experiment was carried out in a "neutral" territory, perhaps the winner mouse might consider the new owner. The male winner could have spent more energy marking the territory when it has a new invader in the territory than the known invader, like the dear Enemy pattern. The loser male could have deposited less urine marks because he was in his territory and the winner male deposited more urine mark in all contexts because he was the owner of the territory. The loser male deposited more urine marks in the presence of the known male because he was not in his territory, so he didn't need to expend energy withdrawing invaders, perhaps acquiring the territory he was lost. If the losing male did not own the territory, the nasty neighbor effect was not applied to this case. It might be a matter of possession of the new territory. In a new study, it should be interesting to see how mice react in a "non-neutral" territory, where one of the mice is already the owner of the territory and the other mice is the invader.

Figure 1 and Figure 4 – Denial of denial (no unfamiliar) in the name of the purple treatment, it is making confusing. You are using "familiar" and "unfamiliar" throughout the text. Change it, please.

Ln 379. When did the experiment take place?

Ln 382 -385 How many generations these two strains (NY2 and NY3) are in captivity?

Ln 558 - Check the references. You missed many journals names abbreviations and dots in the references.

Reviewer #2 (Remarks to the Author):

In this manuscript Miller et al. employ thermal imaging to capture spatiotemporal changes in marking behaviour of male mice. Their findings offer compelling evidence that costly animal signals can be flexibly modulated both in space in time in response to both the competitive context and social environment that an individual experiences. These findings provide exciting validation for thermal imaging as a new approach to measuring scent marking and as such opens new and exciting possibilities for future work on voluntary urination and social signaling.

I found the study to be well executed, the analyses robust and the manuscript to be clearly and comprehensively written. If anything, my main complaint might be that the manuscript is in fact too comprehensive! By this I mean that there is a huge amount of content crammed into a single paper. There are five main figures in the paper, all of which have no fewer than 5 insets – meaning there are a total of at least 25 figures! To the authors' credit, all these figures supplement the main text in a useful way and the manuscript is very clearly written such that they can get away with the inclusion of all this content in a single paper, which is an impressive feat indeed. That being said, it is my opinion that this paper would actually benefit from being split into two separate manuscripts – one on the adjustment of scent marking in response to competition and the other on adjustment of scent marking in response to the familiarity of conspecifics. This feels like a very natural break in the current manuscript and in fact lines 206-214 almost read like a mini-introduction to the second half of the paper. However, as I've said the manuscript is very well written and so such a change need not preclude publication of the paper in its current format. I mainly make this suggestion as I think it would enhance the accessibility and digestibility of the paper(s) without lessening the impact.

Outside of this, I only have one larger concern regarding the study design and some smaller comments for the authors' consideration. My main question regarding study design has to do with the fact that the authors pooled the urine of multiple males for the unfamiliar male treatment. Would the authors expect that pooling urine of multiple males has any unintended effects on scent marking behaviour? That is, are males still likely to perceive this as a single male or would they pick up multiple distinct urinary protein profiles and thus perceive this as multiple unfamiliar males rather than a single unfamiliar individual? If this treatment were being perceived as many unfamiliar individuals could that contribute to the very stark differences observed in how losers vs. winners responded to the presence of unfamiliar conspecifics? Specifically, it might make sense that losers scent mark very little in the presence of multiple potential male aggressors. Maybe there is no chance of this treatment being perceived as multiple different males, but if so, the authors should make this clear to the reader.

Line edits:

L113-114 – Although the number of urine marks detected by thermal imaging and UV imaging did not differ significantly it was still surprising that the UV imaging seemed to show a slightly higher number of urine marks than the thermal imaging given the authors' point that when using UV imaging urine marks deposited in close proximity to other marks will appear as a single deposition. What then is leading to slightly more urine marks under UV imaging? Is this from males tracking urine with paws and tail?

L145-146 – Could the authors explain the patterns in these results a bit more clearly? Given the figure cited is in the SI, it would be helpful to give the reader a bit of a better understanding of HOW spatial signaling is changing without having to go to the supplementary info. This might also be a good place to briefly explain what the regions of interest (ROIs) are and how they are determined (i.e. is this just the half-way point between the wall and barrier)?

L171-172 – So if I understand this correctly, losers who were infrequent markers to begin with, speed up how quickly they scent mark post-fight. Any idea why this might be, given it seems somewhat counterintuitive in light of the other results?

L176-194 – I found this section on the temporal rhythm of urine marking to be the most difficult to follow, which may just be an unfortunate misunderstanding of the terminology the authors use. An "inter-mark interval" (IMI) is simply the length of time between any single urinary event/mark, is that correct? If so, can the authors please make that explicit. I initially thought that what was meant by this was the length of time between urinary "bouts", and I didn't realize this was incorrect until I reached the next paragraph. In L181-183 the authors make the point that they find that the interval between urinary marks to be shorter post-fight than pre-fight and, given this, they classify the type of marking behaviour as "chains" versus "bursts". However, to me this is a bit misleading as the use of "chain" vs. "burst" implies something not only about the rate of scent marking but also about how "clumped" or evenly distributed that marking is (with "bursts" being more clumped, in my mind at least). The authors do elaborate in the next paragraph (L187-194)

on how they classified "marking bouts" and compared the speed of marking within a bout pre- and post-fight. This, to me, better captures the "chain" vs. "burst" idea and so maybe it would be better if the authors introduced these terms at the end of this second paragraph. Additionally, because of the idea that "chain" vs "burst" evokes, I also expected some analysis of the length of the interval between bouts (in addition to the interval between urinary marks within a bout). Is this something the authors looked at? Generally, while this section became clear after several read-throughs, I found it difficult to follow at a first pass and suggest that being more explicit with terminology might help with this.

L221-222 – It seems that ideally the authors would have wanted to expose each individual to the different familiarity treatments to see how within-individual behaviour changed depending on the social environment? But given the experimental design I can see why this was not possible.

L295-296 - I would argue that the authors should not interpret main effects with an interaction term in the model. If the interaction is non-significant, I would remove it and report the main effects from a model without the interactions.

L510-511 – It is still not entirely clear from the methods section how the "regions of interest" were generated. Can the authors provide a bit more detail?

Reviewer #3 (Remarks to the Author):

The authors use a new method of thermal imaging to analyze the spatio-temporal dynamics of male urine marking in wild-derived inbred strains of house mice. Urine marking was documented when a male was exposed to a same-sex competitor (before and after fighting, when separated by a mesh barrier), in an empty arena, and in the same arena with urine from unfamiliar and familiar males. The marked trial was balanced for samples from self, familiar male (=previously encountered) and unfamiliar male (a sample from pooled urine collected from several males of an inbred strain). The method used is innovative since it allows to record (in real time) the onset (latency), time sequence and pattern of urine marking (in bursts or chains), in addition to the location where the urine marks are deposited. It reveals new initial results on signaling behavior that will be interesting for future analysis.

The authors conclude that male signaling behavior (allocation and patterns) varies as a function of initial signaling state (personality differences?), but is additionally modified by experience (losing or winning in an encounter with another male), and when afterwards allowed to countermark urine signals from familiar versus unfamiliar males. This opens the nice potential studying in the future detailed mouse urine marking dynamics, to understand the regulation and adaptive value of signaling behavior in house mice (both in males and in females).

Concerning male (intrasexual) competitive behavior, the results of the study presented are of limited significance. First, signaling in the "marked trials" is compared to signaling in an empty arena (lacking any cues applied in the corners). To allow conclusions on intraspecific communication or competition, however, will require a more appropriate control (urine of sexually immature males, of females, of any other olfactory cue besides urine from sexually mature males), dependent on the specific question to be asked. Second, signaling through urine is fundamentally modified by specific MUPs (major urinary proteins) excreted in the urine, which are up- or down-regulated according to not only own experience and the social situation, but also by own physiological status (health, competitiveness etc.), besides genotype (which is mentioned by the authors). The interpretation of male signaling behavior thus requires incorporating such information, at least by critically discussing such important information on MUPs, which is missing in the ms.

To summarize, the new method and preliminary results presented are valuable for specialists interested in mouse signaling behavior. The authors' general conclusions, on the other hand, are not yet backed enough by empirical data to be of interest for a general audience - to reveal new aspects of either the mechanisms or the adaptive value of animal signaling.

Response to Reviewers: Author responses indicated in blue

Reviewer #1 (Remarks to the Author):

The article is very interesting, relevant, and adequate, but there are some issues we need to discuss and corrections to be made. Your article reaches several audiences by contextualizing the results with neuroscience data. But I have some doubts about the cognition/perception of mice that need to be clarified.

1. The two strains selected are from the same location, this may have interfered with the recognition of familial and unfamiliar individuals. You could have the same results without the fight trial.

Live house mice were collected from the same general geographical area within NY state (near Saratoga Springs). A maximum of two breeding pairs were collected from each site, and all collection sites were at least 500m apart to avoid collecting closely related mice. These wild-caught animals were used to create inbred mouse lines when Sheehan was a postdoc in the Nachman Lab at UC Berkeley. Wild-caught animals were mated, and sibling-sibling pairings were performed since August 2013 to generate inbred lines. More details on the collection methods can be found in Phifer-Rixey et al. 2018¹. The Sheehan Lab received two of these lines in 2016 (which are named in this manuscript "NY2" and "NY3"). The NY2 and NY3 lines were maintained via sibling-sibling pairings in the Sheehan Lab. At the time of experimentation these lines had experienced roughly 12 generations of inbreeding.

Clarification added: In 399-403

To the point that the potential genetic relatedness of these strains may have interfered with the recognition of individuals (related vs. novel): the mice are not closely related (they are likely as or more divergent than 'distantly related' classical inbred strains). Furthermore there is no evidence that mice can detect familial relation via urine profiles except in the instance of an EXACT urine profile match (both in terms of the presence/absence of MUPs, but also the relative proportions of MUPs secreted)²⁻⁴. Prior studies have revealed that house mice use a self-referential MUP-matching system. This includes instances of female inbreeding avoidance³, cooperative nestmate selection⁴, as well as male countermarking to male urine². The mechanism of self vs. non-self marking responses has been further interrogated by minorly adjusting a male's own urine profile⁵. Even slight changes to "self" urine profiles are sufficient to elicit countermarking. Taken together, this suggests that only *very close* genetic relationships can be ascertained prior to meeting that individual (i.e. self-matching mechanism for kin recognition). Non-matched familial and other nest relationships are therefore likely learned through experience with individuals and their unique odors. Additionally, though these mouse strains were collected from the same population they are not related.

Clarification added: In 490-500

Ln 82 – 85:

2. There was just one competitive context (fight trial) between the pairs of male mice, and that made then a familiar pair in your experiment. Why were the mice exposed to only one fight trial?

The reason for a single extended fight trial was multifold. First, we wanted to be certain that within a pair one male clearly lost and the other unequivocally won the contest. We expected the fight trial would need to be extended as all paired males were evenly matched (i.e., age and weight-matched). Most short fight trials (5-15 minutes) performed in other studies involve "skewed" fights, in which one male is intentionally smaller and/or younger to ensure the likelihood of one male winning (most commonly used in intruder-resident assays). With well-matched males, we expected that the resolution of the contests would take longer. Moreover, the size of the arena was considerable larger (dimensions: 50 x 50 cm) than standard house mouse contests (which are typically performed in a cage-sized arena, dimension: 18 x 28 cm). With more space we also expected that some escape and/or avoidance behaviors within the contests would be more accessible, and therefore further extend the time it took to establish dominance within a male pairing.

Second, given that in the first fight we wanted males within each pair to soundly either win or lose the contest, we expected the encounter would be both physically exhausting and stressful. We did not want to over-stress the males, and thus limited the full physical contest to a single encounter. In preliminary testing, we attempted an alternative contest design involving a short series of contest encounters. However, the combination of added handling, disturbance, and repeated contests appeared to add undue stress to the mice. Because of this, we opted for a single lengthier contest for the purposes of this study and the health of the subjects.

Third, while the 2nd Mesh trial (on the 3rd trial day) didn't allow for complete physical contact (aside from mild contact through the mesh barrier), competitors could see, smell, and hear their competitor for 30 minutes. One of the goals of the paired-series design of the experiment was to reveal that their competitors were still present in the environment, while minimizing the physical stressors of the fight itself.

Clarification added: In 456-459

3. What were the criteria for defining that pair of male mice are familiar to each other? As nothing about how animals are housed was mentioned in your article, it might be interesting to bring more details about this.

The criteria for familiarity: familiar males had to have physical exposure with a previously novel individual for an extended period of time (1 hour on their first day of exposure). On the first trial day males were exposed to a 30-minute Mesh trial, followed by a 30-minute Fight trial with their paired competitor. Each male also entered into a 2nd Mesh trial on the 3rd trial day with their same male competitor. Therefore, prior to the urine-marked trial on the 4th and final day of the trial series each male spent a total of 1.5 hours with their “familiar” competitor, and had an established dominance relationship.

Clarification added: In 222-225

Housing details can be found in the methods section:

- **Ln 408-413:** “At weaning age (3-4 weeks) males were placed into a holding cage alone for 1-2 weeks, and were subsequently paired with a female to allow for sexual experience, as sexually naïve mice are known to exhibit different social behaviors⁶. All males were allowed to reach adulthood (3-5 months old by the time of experimental testing) and had the opportunity to produce one or more litters. All holding and breeding cages contained corn cob bedding, cardboard huts, and cotton nestlets. Mice were maintained in an Animal Care facility at Cornell University with a 14:10 shifted light:dark cycle (dark period: 12PM-10 PM) and were provided food and water *ad libitum*.”
- **Ln 420-421:** All males were in breeding cages at the time of the experiment and most successfully reproduced (84%) prior to start of the trial series.”

Ln 382 – 383:

4. Why did you test only in two strains?

The reason for using only two strains was largely logistical. This experiment was already a considerable undertaking in terms of the number of experimental animals and the stimulus controls across treatment groups, from a mouse care management perspective. With ample time and funding, incorporating more strains would have been interesting. It would be particularly intriguing to perform a round-robin series of competitions across 3 or more strains, and examine whether win-win, win-lose, lose-lose, lose-win scenarios yield different competitive marking responses.

5. Did the effect hold in a third strain of mice compared to these two (NY2 and NY3)?

The mentioned line numbers (382-383) are in the methods section, so it's unclear which effect is being referring to here, though contextually we are guessing you're referring to the marking responses toward familiar vs. unfamiliar urine. We did not perform this trial series on a third strain of mouse. Given that we observed no strain-specific effects on marking behavior, we expect the competitive marking responses would hold across other strains of mice as well.

6. Are the urinary proteins profiles of NY2 and NY3 more similar with each other than are they similar with the C57BL/6 urinary protein profile?

This concern is irrelevant for the current study because house mice recognize differences in urinary protein profiles based on just a single MUP difference as well as changes in relative ratios of existing MUPs. It's clear from the data that NY2 and NY3 mice distinguish between their own urine and that of their competitor. From a sensory ecology standpoint, all three strains have distinct urinary profiles. Further details on the MUP expression of the NY2 and NY3 males can be found in Sheehan et al. 2019⁷. Critically, the absolute number of central MUPs expressed and secreted by individuals has not been shown to have any effect on male marking behavior, scent investigation, female mate preference, or female cooperative nesting²⁻⁴. In each instance, house mice have been found to use *self-referential matching*. Self vs. non-self MUP profiles are the means by which familial relationship can be ascertained without prior experience with an individual. In other words, mice do not respond to a gradient of familiarity in relation to their own MUP type, only whether it's an exact match to their own profile or not.

Prior research has found that even minor differences in MUP profile are perceived as distinct from self urine²⁻⁴. One such study found that manipulating *just* the relative proportion of a *single* MUP already present in a male's MUP profile was sufficient to elicit strong countermarking responses (as though to a novel male)⁵. In a recent paper, we similarly find that females also respond to even slight proportional shifts in MUP profiles⁸ as distinct identities. In this study, the “familiar male” urine stimulus contains a MUP profile that is considerably more distinct than just a proportional shift of a protein within the MUP profile. The familiar male urine has both presence/absence and proportional differences from the focal male. Given this, the fact that winners *don't* mark highly to familiar male urine is striking (Figure 4A-C). Prior to this study, research would suggest that males should countermark indiscriminately to the urine of other males (i.e. non-self urine). This study is unique in that we presented urine stimuli from a male with which the focal male has an established relationship, as opposed to the urine being novel (either from a novel individual or novel via profile manipulation). It's arguably even more surprising if the winning males do in fact perceive the arena as an extension of his territory (as discussed in comment #8). This would suggest the urine of another male (familiar or not) is an explicit intrusion on their territory by a male they recently defeated, and yet the winning males do not mark highly to this urine.

With losing males, we face some challenges in seeing significant differences across treatments as there is an overall floor effect. Despite the fact that losers are simply marking lowly overall, we observe trends among losers that indicate differential responses to self and familiar male urine. For example, losers tend to mark more to the S-FM treatment relative to the S-S treatment and compared to all other treatments as well (Figure 4A). Losers also have a second peak in their temporal marking distributions at approximately 600s in response to the S-FM treatment. This is not observable to the S-S treatment which has a single peak with a slow decline. Furthermore, losers appear to have the most detectable burst-like marking bouts in the S-FM trials, though the low levels of marking overall do not reveal this as a significant pattern (Figure S6D).

Furthermore, any possibility of the similarity vs. difference in MUP type cannot explain the marking patterns observed. We find that winners and losers respond very differently to the urine of a novel male. The extent of the novelty could in theory determine the extent of the effect, but the urine is unquestionably that of a novel male for both groups.

Addressed with added clarification added: In 222-228, 433-438, 490-500, 511-524

7. Two strains from the same site (Saratoga Springs) were selected to be familiar pair and a different strain was selected from another site to be an unfamiliar (C57BL/6), it is not possible to say whether the effect was due to genetic differences in recognition (differences in protein profiles) or due to a difference in experimental treatment. Mice could respond differently to more similar and more different genetic profiles (urinary protein profile), and your experiment did not allow for that distinction.

This question seems to stem from an unintentionally confusing wording on our part about the origin of these strains, which we have now amended. The mice that gave rise to each strain came from a large wild population with high levels of genetic diversity. There is substantial genetic diversity within wild house mouse populations collected in a single metropolitan area compared to variation present among all the classic inbred strains (e.g. Laurie et al 2007 PLoS Genetics)⁹. It is true that their ancestors were collected in the vicinity of Saratoga Springs NY, but the two strains originate from distinct geographical sites. The mouse strains are *not* closely related to each other. Moreover, sequencing done by colleagues shows they differ by *millions* of SNPs (personal communication from Beth Dumont, Jax Lab). The differences between these two NY strains are *much* greater than between C57 and other classic inbred strains such as CH3 or DBA.

In addition to the fact that these strains are not close relatives, the claim that the experiment would not show recognition if relatedness was an issue (which it is not) is illogical in itself. The experiment shows that mice behave differently depending on which urine is present. What's more the "outgroup" urine from C57 is clearly treated as a competitor stimulus by the winning males. The novel finding from this study is that mice treat the **same exact stimulus** differently depending on their recent fight outcome. Additionally, the response to the S-FM treatment shows an increase from baseline for losers, but no change from baseline for winners. Regardless of what we call the discrimination of urine from different strains by the subjects, it is clear that (1) mice are discriminating among urine types and (2) winning or losing causes different responses to the same stimulus.

Prior research shows even very slight differences in urine profile from a male's own urine is sufficient to elicit marking behavior. Several studies find that mice use a self-referential matching mechanism (i.e. have the exact same MUP profile). One of the first studies to examine this in detail (Hurst et al. 2001) presented adult males with 4 different urine stimuli: (1) self urine, (2) novel male urine, (3) sibling male urine with the same MUP profile, and (4) sibling male urine with a different MUP profile. Scent marking responses were examined. Hurst et al. found that males scent mark comparably to (1) self urine and (3) same-MUP male sibling urine, and that males mark significantly more to (2) novel male and (4) different-MUP male sibling urine. What's particularly striking here is that all siblings are familiar, but are treated very differently depending on whether the MUP profile is an exact MUP match or not.

This self vs. non-self distinction has been further explored with careful manipulation of urine profiles⁵. This study found that adding an additional MUP, or adjusting the relative proportions of existing MUPs, was sufficient to alter the perceived urine identity and stimulate countermarking behavior as if toward a novel male. Strikingly, in this same study the researchers manipulated a male's home cage odor by consistently adding MUPs they would not otherwise produce. By chronically exposing males to an additional MUP, they "tricked" these males into thinking that this additional MUP was present in their "self" profile. When presented with their own (unmanipulated) urine, these males countermarked as if to a novel male. This indicates that minor differences are ecologically relevant to house mice, and that immediate odor experiences in their environment are important for creating and maintain odor signatures, including the "self" signature. We similarly find in our recent study that very minor alternations in familiar male urine causes females to respond as if the stimuli are novel males, across both estrus and pregnancy⁸.

In the current study, all males prior to the experiment were exposed solely to their same strain, and therefore only had experience their own strains' MUP profile. The exposure to a new strain with a distinct MUP profile (regardless of how similar or different the profiles were) would have been entirely novel, and would have no means for males to ascertain relatedness. Accordingly, it seems highly unlikely that similarity/difference in profile would govern male responses to the familiar vs. unfamiliar urine stimuli presented in this study. Particularly as recent competitive experience appears to have strong effects on marking behavior, and odor memory signatures are updated with experience^{5,10,11}

Addressed with added clarification added: In 222-228, 339-403, 433-438, 490-500, 511-524

8. Ln 343 – 351 Your conclusion did not seem to consider that the disputed territory had a winner, and that winner could be the owner of the territory. As the experiment was carried out in a “neutral” territory, perhaps the winner mouse might consider the new owner. The male winner could have spent more energy marking the territory when it has a new invader in the territory than the known invader, like the dear Enemy pattern. The loser male could have deposited less urine marks because he was in his territory and the winner male deposited more urine mark in all contexts because he was the owner of the territory. The loser male deposited more urine marks in the presence of the known male because he was not in his territory, so he didn't need to expend energy withdrawing invaders, perhaps acquiring the territory he was lost. If the losing male did not own the territory, the nasty neighbor effect was not applied to this case. It might be a matter of possession of the new territory. In a new study, it should be interesting to see how mice react in a “non-neutral” territory, where one of the mice is already the owner of the territory and the other mice is the invader.

This is a very valid point that was not sufficiently addressed in the manuscript. If I'm understanding the comment correctly, your concern is that after the first trial day the arena may no longer be “neutral” territory. Because of this, how winning and losing males perceive the space of the arena itself may affect the observed marking patterns. While there is a clear divergence in response toward familiarity among winning and losing males, the perspective raised rightly calls out the need for added nuance when interpreting the results of this study.

All males (winners and losers) have a home cage territory that they share with a breeding female, so in that sense are “territory owners”. Though importantly, the arena is not their home territory (i.e. their home cage). As a result, the arena could be perceived as (1) an extension of their territory, (2) as a disputed region, or (3) as neutral area (i.e., overlapping home ranges). The design of this study doesn't allow for a clear statement about these possible perceived differences. This experimental design was selected to allow for careful control of the odor environment and the placement of urine stimuli. However, future work examining such social odor stimuli in the home cages of competitors would be a great way to investigate how males might respond differently to an explicit “intruder” relative to a “familiar competitor”.

What this study does find is that having an established competitive relationship with another male strongly affects signaling dynamics. Given that males aren't meeting in their actual home cage territories, and they don't have established territory boundaries with each other, the dear enemy and nasty neighbor effects are not proven here. Nevertheless, the framework presented by these effects is still helpful to consider given the differences observed toward familiar competitors.

Clarification added: In 360-365

9. Figure 1 and Figure 4 – Denial of denial (no unfamiliar) in the name of the purple treatment, it is making confusing. You are using “familiar” and “unfamiliar” throughout the text. Change it, please.

I acknowledge that the terminology can be cumbersome. As mentioned in the text, we initially intended to compare within-arena scent mark responses, but the space was not sufficient to do so (In 243-246, 559-566). As a result, we combined treatments because the marking patterns were most strongly affected by the presence/absence of unfamiliar male urine in the environment, and to improve statistical power. In prior versions of this manuscript we did actually use “familiar” vs. “unfamiliar” terminology throughout. However, other colleagues requested it be changed to “no unfamiliar” to improve the accuracy of the treatment groups and analyses given that the S-S and S-FM treatments are collapsed together. This appears to be a point of disagreement across different readers, so for the moment we have kept the “no unfamiliar” and “unfamiliar” labels in the figures.

10. Ln 379. When did the experiment take place?

Trials were performed between the months of May - November during the years of 2018 and 2019. The winter months were avoided to prevent strong seasonal affects in which the mice are less active. Laboratory mice exhibit seasonal variation with respect to certain physiological parameters like serum concentrations of sex hormones, suggesting a possible mechanism for the internalization of annual time independent of light cycle, temperature and humidity¹². While the available literature provides conflicting evidence as to whether these effects extend to behavior, we nonetheless took measures to avoid such confounds¹³, particularly as the mice were wild-derived and recently inbred from the northeastern US. The start dates for each trials series are also provided in the supplemental data sheet “summary_thermal.marks.space.use.CSV” in the data column “StartDate”

Clarification added: In 423-430

11. Ln 382 -385 How many generations these two strains (NY2 and NY3) are in captivity?

The two lines have been maintained by sibling-sibling mating since August 2013. At the time of trials in 2017 they experienced roughly 12 generations of inbreeding.

Clarification added: In 399-403

12. Ln 558 - Check the references. You missed many journals names abbreviations and dots in the references.

Thank you! They have been updated.

Reviewer #2 (Remarks to the Author):

In this manuscript Miller et al. employ thermal imaging to capture spatiotemporal changes in marking behaviour of male mice. Their findings offer compelling evidence that costly animal signals can be flexibly modulated both in space in time in response to both the competitive context and social environment that an individual experiences. These findings provide exciting validation for thermal imaging as a new approach to measuring scent marking and as such opens new and exciting possibilities for future work on voluntary urination and social signaling.

I found the study to be well executed, the analyses robust and the manuscript to be clearly and comprehensively written. If anything, my main complaint might be that the manuscript is in fact too comprehensive! By this I mean that there is a huge amount of content crammed into a single paper. There are five main figures in the paper, all of which have no fewer than 5 insets – meaning there are a total of at least 25 figures! To the authors' credit, all these figures supplement the main text in a useful way and the manuscript is very clearly written such that they can get away with the inclusion of all this content in a single paper, which is an impressive feat indeed. That being said, it is my opinion that this paper would actually benefit from being split into two separate manuscripts – one on the adjustment of scent marking in response to competition and the other on adjustment of scent marking in response to the familiarity of conspecifics. This feels like a very natural break in the current manuscript and in fact lines 206-214 almost read like a mini-introduction to the second half of the paper. However, as I've said the manuscript is very well written and so such a change need not preclude publication of the paper in its current format. I mainly make this suggestion as I think it would enhance the accessibility and digestibility of the paper(s) without lessening the impact.

Outside of this, I only have one larger concern regarding the study design and some smaller comments for the authors' consideration.

13. My main question regarding study design has to do with the fact that the authors pooled the urine of multiple males for the unfamiliar male treatment. Would the authors expect that pooling urine of multiple males has any unintended effects on scent marking behaviour? That is, are males still likely to perceive this as a single male or would they pick up multiple distinct urinary protein profiles and thus perceive this as multiple unfamiliar males rather than a single unfamiliar individual? If this treatment were being perceived as many unfamiliar individuals could that contribute to the very stark differences observed in how losers vs. winners responded to the presence of unfamiliar conspecifics? Specifically, it might make sense that losers scent mark very little in the presence of multiple potential male aggressors. Maybe there is no chance of this treatment being perceived as multiple different males, but if so, the authors should make this clear to the reader.

Multiple studies have shown that mice respond using a self-referential MUP matching system²⁻⁵. If the MUP profile matches their own, males do not countermark to it (i.e. it's treated as "self" urine). A study further found that males cannot distinguish within-strain urine apart¹⁴. Because males of the same strain are inbred they have the same exact MUP profile and smell like identical scent twins. Pooling the urine of individuals of the same strain therefore does not affect the MUP profile, as all inbred individuals share the same exact profile¹⁴.

However, while the MUP profiles do not vary across individuals from a given strain, the overall levels of MUP production¹⁵⁻¹⁷ (high vs. low) do vary from male-to-male as do various volatiles (e.g. steroid byproducts)¹⁸⁻²⁰. We pooled the C57 ("unfamiliar") male urine to ensure that the urine stimuli presented to males was *exactly the same* across all treatments and trials without any individual-specific effects (e.g., differences in steroid biproducts secreted in urine). Therefore, all experimental males were exposed to the *exact same* "unfamiliar male" urine stimulus, with the same MUP profile, the same MUP levels, and the same blend of volatiles.

Finally, from a mechanistic standpoint it is unclear that it would even be possible for mice to distinguish a urine mark as belonging to one individual or the pooled urine of multiple individuals, especially if all those individuals are unfamiliar. It is not clear how or whether the sensory system distinguishes which elements of the overall scent profile were contributed by which individual in the case of pooled and homogenized urine. Whether from a single individual or pooled from many, the urine would have a single cohesive odor mixture that with a given combination of volatile and non-volatile elements that would be consistent within a urine treatment.

Clarification added: In 226-228, 490-500, 511-524

Line edits:

14. L113-114 – Although the number of urine marks detected by thermal imaging and UV imaging did not differ significantly it was still surprising that the UV imaging seemed to show a slightly higher number of urine marks than the thermal imaging given the authors' point that when using UV imaging urine marks deposited in close proximity to other marks will appear as a single deposition. What then is leading to slightly more urine marks under UV imaging? Is this from males tracking urine with paws and tail?

It was a bit of a surprise to us as well. We originally expected that thermal imaging would detect more marks overall. However, the key difference between the imaging methods is that UV imaging cannot distinguish between *deposition* and *distribution* events, while thermal imaging can (In 110-113). This appears to be in large part due to males distributing urine by tracking it around the arena after walking through it with their paws and tail. This was observed as males will distribute very small "cool

spots" after walking through a large urine mark that was previously deposited. These distribution events are never observed as "hot spots", and are thus not counted as a deposition event in the thermal datasets. We also observed instances in which males overmarked the same spot. Overmarking events are only detectable with thermal imaging, as under UV imaging the urine mark would have bled together to appear as a single mark. In terms of the relative differences in marks detected by both methods, UV imaging does trend towards detecting more marks overall (Figures 1G & Figure S2). This indicates that the "distribution effect" is more prominent than the "overmarking effect" on mark detection methods. Future studies could easily examine mark distribution events as well using the thermal imaging method. Not surprisingly, when males are placed in an arena with marks already in the environment (urine-marked trials), the likelihood of overmarking is higher. This is reflected in the fact that the number of marks detected tended to be higher for UV imaging for the Mesh1, Empty and Mesh2 trials, while thermal recording detected more marks in the Marked trials (Figure S2).

Clarification added: In 110-113, 543-553

15. L145-146 – Could the authors explain the patterns in these results a bit more clearly? Given the figure cited is in the SI, it would be helpful to give the reader a bit of a better understanding of HOW spatial signaling is changing without having to go to the supplementary info. This might also be a good place to briefly explain what the regions of interest (ROIs) are and how they are determined (i.e. is this just the half-way point between the wall and barrier)?

Descriptions of the mesh trial regions of interest (ROIs) used in spatial analyses have been added. This section is kept brief because the spatial differences were overall mild or non-existent. We suspect this was in part due to the size of the arena. In a larger space we expect that clear spatial patterns would emerge.

Clarification added: In 143-145, 555-557

16. L171-172 – So if I understand this correctly, losers who were infrequent markers to begin with, speed up how quickly they scent mark post-fight. Any idea why this might be, given it seems somewhat counterintuitive in light of the other results?

Interestingly, there actually appears to be an equalizing effect on the latency to mark after losing a fight: high-marking individuals slow down and low-marking individuals speed up, such that all losing males end up marking at the same speed post-fight. Initially, this result surprised us as well. However, as we examined more temporal marking features there appear to be important and generalized priming effects that occur on the timing of urine marking in response to competition. In other words, experiencing competition pushes all males to mark more rapidly in the environment. Such that while losers mark lowly, they are still primed to mark relatively quickly (though not as quickly as winners). This is addressed currently in the discussion (In 368-375) but some interpretation has been added to this results section as well.

Clarification added: In 173-178

17. L176-194 – I found this section on the temporal rhythm of urine marking to be the most difficult to follow, which may just be an unfortunate misunderstanding of the terminology the authors use. **An "inter-mark interval" (IMI) is simply the length of time between any single urinary event/mark, is that correct? If so, can the authors please make that explicit.** I initially thought that what was meant by this was the length of time between urinary "bouts", and I didn't realize this was incorrect until I reached the next paragraph.

Yes, you are correct the "inter-mark interval" (IMI) is the time between two mark events. However, the IMI is largely used to examine the differences in time between marking events in the context of marks that occur in series together (i.e. mark bouts). Clarification has been added to the paragraph introducing this terminology in the results section.

Clarification added: In 181-185

18. In L181-183 the authors make the point that they find that the interval between urinary marks to be shorter post-fight than pre-fight and, given this, they classify the type of marking behaviour as "chains" versus "bursts". However, to me this is a bit misleading as the use of "chain" vs. "burst" implies something not only about the rate of scent marking but also about how "clumped" or evenly distributed that marking is (with "bursts" being more clumped, in my mind at least). The authors do elaborate in the next paragraph (L187-194) on how they classified "marking bouts" and compared the speed of marking within a bout pre- and post-fight. This, to me, better captures the "chain" vs. "burst" idea and so maybe it would be better if the authors introduced these terms at the end of this second paragraph. Additionally, because of the idea that "chain" vs "burst" evokes, I also expected some analysis of the length of the interval between bouts (in addition to the interval between urinary marks within a bout). Is this something the authors looked at? Generally, while this section became clear after several read-throughs, I found it difficult to follow at a first pass and suggest that being more explicit with terminology might help with this.

We have removed the terminology mark "chains" and "bouts" in this paragraph (In 182-193), but introduce them later as suggested when discussing mark bouts specifically.

Clarification added: In 196-198

19. L221-222 – It seems that ideally the authors would have wanted to expose each individual to the different familiarity treatments to see how within-individual behaviour changed depending on the social environment? But given the experimental design I can see why this was not possible.

Indeed, the original goal was to compare within-treatment responses (e.g. how a male responded to self urine relative to familiar male urine within an environment). Unfortunately, the space was simply not sufficient to distinguish within-treatment responses to different urine stimuli. In preliminary testing it was hard to detect this particular challenge given the inherent variation in urine marking behavior, as while scoring the trials it became very clear that mark sequences extended across multiple ROIs. Future studies testing similar social odors would benefit greatly from utilizing a larger space.

20. L295-296 - I would argue that the authors should not interpret main effects with an interaction term in the model. If the interaction is non-significant, I would remove it and report the main effects from a model without the interactions.

While the interaction term isn't significant it is modest ($p = 0.08$). Including the interaction term further allowed us to perform post hoc group comparisons. Removing the interaction term results a similar overall pattern of significance, however the stimulus treatment has a stronger effect ($p = 1.223e-14$) and fight outcome has a more mild effect ($p = 0.077$). Given this, we argue that it's reasonable to keep the interaction term. Though we did realize that the supplementary statistics table for this model in the previous manuscript was incorrect (if this was the discrepancy being addressed in your comment). This has been fixed, and all supplementary statistics tables have been updated to correctly correspond to all in-text statistics.

21. L510-511 – It is still not entirely clear from the methods section how the “regions of interest” were generated. Can the authors provide a bit more detail?

Additional details on the generation of the ROIs have been added to the methods in both the “behavioral scoring and analysis” section and the “tracking” section.

Clarification added: In 555-559, 573-578

Reviewer #3 (Remarks to the Author):

The authors use a new method of thermal imaging to analyze the spatio-temporal dynamics of male urine marking in wild-derived inbred strains of house mice. Urine marking was documented when a male was exposed to a same-sex competitor (before and after fighting, when separated by a mesh barrier), in an empty arena, and in the same arena with urine from unfamiliar and familiar males. The marked trial was balanced for samples from self, familiar male (=previously encountered) and unfamiliar male (a sample from pooled urine collected from several males of an inbred strain). The method used is innovative since it allows to record (in real time) the onset (latency), time sequence and pattern of urine marking (in bursts or chains), in addition to the location where the urine marks are deposited. It reveals new initial results on signaling behavior that will be interesting for future analysis.

The authors conclude that male signaling behavior (allocation and patterns) varies as a function of initial signaling state (personality differences?), but is additionally modified by experience (losing or winning in an encounter with another male), and when afterwards allowed to countermark urine signals from familiar versus unfamiliar males. This opens the nice potential studying in the future detailed mouse urine marking dynamics, to understand the regulation and adaptive value of signaling behavior in house mice (both in males and in females).

22. Concerning male (intrasexual) competitive behavior, the results of the study presented are of limited significance. First, signaling in the "marked trials" is compared to signaling in an empty arena (lacking any cues applied in the corners). To allow conclusions on intraspecific communication or competition, however, will require a more appropriate control (urine of sexually immature males, of females, of any other olfactory cue besides urine from sexually mature males), dependent on the specific question to be asked.

As indicated in the manuscript (In 243-246, 559-566) we initially planned to compare the scent marked corners within the "marked trials." However, in analyzing the movements and marking patterns of the mice we determined that the space was simply too small to detect such differences. Given the variability inherent to scent marking, this was challenging to troubleshoot in preliminary work. Instead we treated the "marked trials" as an entire social scent environment (In 245-249), for which we find that the presence vs. absence of unfamiliar male urine in that environment has clear effects on marking behavior. Comparing marking patterns in the empty trials to the urine-marked trials provided a control for marking levels for each male in the absence of a live competitor or scent marks in the environment. While mice mark less overall in the empty arenas, males still mark predictably based on both their initial mark investment and whether they won or lost (In 232-233) This provided a clear baseline marking level for each male in the absence of a conspecific or urine in the environment post-fight.

We appreciate the interest you express in exploring how different social signals may further modulate scent marking behavior. This is a rich area for future studies to investigate given the tractability of the thermal recording method present here. Including additional stimuli within this study would have lengthened the manuscript considerably, and would not have been logistically possible given time and financial constraints. We are in agreement with reviewer #2, that in many ways this manuscript could be split up into two papers given its length and the extensive analyses present within it. Fortunately, this study builds upon a rich existing literature of house mouse scent mark signaling. In which, many of the social signals you listed have been previously well examined. The goals of this study (In 76-78) were to: "(1) implement thermal recording as a method for measuring scent marking in social contexts, (2) examine how competitive experience alters marking behavior, and (3) test the hypothesis that familiarity is important for signal allocation decisions". The thermal recording method allowed us to explore temporal axes of variation, and thus examine scent marking dynamics in greater depth than previous studies.

23. Second, signaling through urine is fundamentally modified by specific MUPs (major urinary proteins) excreted in the urine, which are up- or down-regulated according to not only own experience and the social situation, but also by own physiological status (health, competitiveness etc.), besides genotype (which is mentioned by the authors). The interpretation of male signaling behavior thus requires incorporating such information, at least by critically discussing such important information on MUPs, which is missing in the ms.

Indeed, the overall level of MUP production does change with a number of social and physiological factors in house mice. MUP production is testosterone and growth hormone dependent, mice start secreting MUPs when they enter adulthood²¹. Adult male mice tend to secrete higher levels of MUPs on average than adult female mice, though these patterns are highly variable in the wild^{22,23}. Higher total MUP levels have been shown to be associated with dominance^{15,16,24}. Additionally, MUP levels tend to be higher in breeding males with territories¹⁶.

In our experimental design we accounted for the age, reproductive state, and territorial status of individuals prior to the start of the trial series each male was exposed to. All experimental males are fully mature adults (3-5 months old, have their own territory (i.e., home breeding cage) that they share with a breeding female, have sexual experience, and almost all males (84%) had successfully reproduced by the start of the trial series (In 408-411, 420-421). Males are also age and size-matched to their paired competitors (In 419-420). Importantly, it takes at least 2 weeks to observe changes in MUP levels in response to social competition or dominance hierarchy shifts^{15,25}. Given the 2-week long timeline for detecting changes in MUP investment and concentration in urine post social encounters, we would not have been able to observe such differences within our much shortened experimental timeline of 4 days. Initially we still planned to collect urine for protein analyses pre- and post-trial series for the goal of examining whether initial protein levels were at all predictive of fight outcome. However, it was very challenging to collect enough urine to present territory-like stimuli in the urine-marked trials and have enough urine for protein analyses as well. Particularly, as we collected urine only 1-week prior to minimize any temporal differences (age, state, etc).

Because of these collection restraints, we prioritized using urine as stimuli for our experiments, given the age, state, weight and food controls placed on all experimental males prior to testing. It would be interesting in future studies to examine the urinary protein levels of males pre and post competitive encounters. This would likely require different protocols for urine collection or urine stimulus presentations.

We also accounted for individual-specific MUP concentrations in our “unfamiliar male” urine stimulus. By pooling urine collected from males of the strain, the pooled urine would contain the same concentration of MUPs. Thus all males were exposed to the exact same unfamiliar male stimulus.

Clarification added: In 518-524

24. To summarize, the new method and preliminary results presented are valuable for specialists interested in mouse signaling behavior. The authors' general conclusions, on the other hand, are not yet backed enough by empirical data to be of interest for a general audience - to reveal new aspects of either the mechanisms or the adaptive value of animal signaling.

This study provides evidence that males implement identity information in their signaling decisions, and will flexibly adjust their signaling output accordingly. While recognition systems and signal investment have been well studied across many species and systems, dynamic shifts in signaling effort are under-studied. This study highlights the importance of examining dynamic signals, and considering time as an important axis of variation in signaling behavior.

We further identify a cohort of low-marking competitive males that have previously never been identified, highlighting the possibility of alternative signaling strategies in house mice. The possibility of a scent-“silent” strategy in mammals has not been explored, yet seems highly plausible given the time and energetic costs involved. Moreover, the use of thermal imaging has revealed an entirely new axis of scent mark signaling features which has been under-explored, not just in mice but all scent marking species. All of these points are of relevance to the research communities of animal communication and signaling.

Your point that this research is relevant to the mouse research community is absolutely correct. Mouse social behaviors are used across a massive swath of neuroscience and medical research studies. This study provides an ecologically relevant social behavior that can be examined with fine spatial and temporal resolution, and thus can be paired with neural recordings or genetic studies. For example, understanding the social odor landscape and behavioral responses to it would be incredibly useful to hippocampal researchers interested in social memory. The rapid temporal shifts in marking responses toward competitive experience are relevant to all researchers interested in aggression, winner-loser effects, and status. The fine-scale temporal information on urine deposition is valuable to researchers interested in the neurophysiological underpinnings of voluntary urination.

REFERENCES

1. Phifer-Rixey, M. *et al.* The genomic basis of environmental adaptation in house mice. *PLOS Genetics* **14**, 1–29 (2018).
2. Hurst, J. L. *et al.* Individual recognition in mice mediated by major urinary proteins. *Nature* **414**, 631–634 (2001).
3. Sherborne, A. L. *et al.* The genetic basis of inbreeding avoidance in house mice. *Current Biology* **17**, 2061–2066 (2007).
4. Green, J. P. *et al.* The genetic basis of kin recognition in a cooperatively breeding mammal. *Current Biology* **25**, 2631–2641 (2015).
5. Kaur, A. W. *et al.* Murine pheromone proteins constitute a context-dependent combinatorial code governing multiple social behaviors. *Cell* **157**, 676–688 (2014).
6. Stowers, L. & Liberles, S. D. State-dependent responses to sex pheromones in mouse. *Current Opinion in Neurobiology* **38**, 74–79 (2016).
7. Sheehan, M. J., Campbell, P. & Miller, C. H. Evolutionary patterns of major urinary protein scent signals in house mice and relatives. *Molecular Ecology* **28**, 3587–3601 (2019).
8. Miller, C. H. *et al.* Reproductive state switches the valence of male urinary pheromones in female mice. *bioRxiv* (2022) doi:10.1101/2022.08.22.504866.
9. Laurie, C. C. *et al.* Linkage Disequilibrium in Wild Mice. *PLoS Genet* **3**, e144 (2007).
10. Roberts, S. A., Davidson, A. J., Beynon, R. J. & Hurst, J. L. Female attraction to male scent and associative learning: the house mouse as a mammalian model. *Animal Behaviour* **97**, 313–321 (2014).
11. Roberts, S. A. *et al.* Individual odour signatures that mice learn are shaped by involatile major urinary proteins (MUPs). *BMC Biol* **16**, 48 (2018).
12. Mock, E. J., Kamel, F., Wright, W. W. & Frankel, A. I. Seasonal rhythm in plasma testosterone and luteinising hormone of the male laboratory rat. *Nature* **256**, 61–63 (1975).
13. Ferguson, S. A. & Maier, K. L. A review of seasonal/circannual effects of laboratory rodent behavior. *Physiology & Behavior* **119**, 130–136 (2013).
14. Nevison, C. M., Barnard, C. J., Beynon, R. J. & Hurst, J. L. The consequences of inbreeding for recognizing competitors. *Proceedings of the Royal Society of London: Series B* **267**, 687–694 (2000).
15. Lee, W., Khan, A. & Curley, J. P. Major urinary protein levels are associated with social status and context in mouse social hierarchies. *Proceedings of the Royal Society B* **284**, 20171570 (2017).
16. Garratt, M. *et al.* Is oxidative stress a physiological cost of reproduction? An experimental test in house mice. *Proceedings of the Royal Society B* **278**, 1098–1106 (2011).
17. Janotova, K. & Stopka, P. The level of major urinary proteins is socially regulated in wild *Mus musculus musculus*. *Journal of Chemical Ecology* **37**, 647–656 (2011).
18. Novotny, M., Harvey, S. & Jemiolo, B. Chemistry of male dominance in the house mouse, *Mus domesticus*. *Experientia* **46**, 109–113 (1990).
19. Harvey, S., Jemiolo, B. & Novotny, M. Pattern of volatile compounds in dominant and subordinate male mouse urine. *Journal of Chemical Ecology* **15**, 2061–2072 (1989).
20. Apps, P. J., Rasa, A. & Viljoen, H. W. Quantitative chromatographic profiling of odours associated with dominance in male laboratory mice. *Aggressive Behavior* **14**, 451–461 (1988).
21. Hurst, J. L. & Beynon, R. J. Scent wars: the chemobiology of competitive signalling in mice. *BioEssays* **26**, 1288–1298 (2004).
22. Stopka, P., Janotova, K. & Heyrovsky, D. The advertisement role of major urinary proteins in mice. *Physiology & Behavior* **91**, 667–670 (2007).
23. Cheetham, S. A. *et al.* The genetic basis of individual-recognition signals in the mouse. *Current Biology* **17**, 1771–1777 (2007).
24. Guo, H., Fang, Q., Huo, Y., Zhang, Y. & Zhang, J. Social dominance-related major urinary proteins and the regulatory mechanism in mice. *Integrative Zoology* **10**, 543–554 (2015).
25. Garratt, M. *et al.* Tissue-dependent changes in oxidative damage with male reproductive effort in house mice. *Functional Ecology* **26**, 423–433 (2012).

REVIEWERS' COMMENTS:

Reviewer #1 (Remarks to the Author):

Thanks to Miller and collaborators for responding to queries and for modifications to the manuscript. The manuscript is suitable for publication.

Reviewer #2 (Remarks to the Author):

The authors have done an excellent job responding to the reviewer comments. I found the study to be carefully executed, the statistical analyses appropriate and the manuscript extremely well written. The authors have done an admirable job of conveying an enormous amount of information in a condensed space. I strongly believe this manuscript will be an important addition to the literature on social signaling behaviour. I have no further suggestions and am happy to recommend this paper for publication.

Reviewer #3 (Remarks to the Author):

I am happy with the authors' responses to my comments, and with the clarifications added to the ms.

I only have very few minor comments:

line 289: add info that urine had been collected (appr. one week?) before the onset of the experiment. Given your results, its important to note here that urine had been collected before males were exposed to a competitor.

line 385/86: add "of" "... aspects of marking behaviour ..."

Line 640: delete "apart" (or: "... cannot tell ... apart by ...")

line 699: delete empty space before comma